# The cation channel mechanisms of subthreshold inward depolarizing currents in the mice VTA dopaminergic neurons and their roles in the chronic-stress-induced depression-like behavior

Jing Wang[1,2,3,4], Min Su[1,5], Dongmei Zhang[1,6], Ludi Zhang[1], Chenxu Niu[1], Chaoyi Li[1], Shuangzhu You[1], Yuqi Sang[1,7,8], Yongxue Zhang[1,9], Xiaona Du[1], Hailin Zhang[1,10]*

[1]Department of Pharmacology, Hebei Medical University, Shijiazhuang, China; [2]Department of Chinese Medicinal Chemistry, Hebei University of Chinese Medicine, Shijiazhuang, China; [3]The Key Laboratory of Neural and Vascular Biology, Ministry of Education, Hebei Medical University, Shijiazhuang, China; [4]The Key Laboratory of New Drug Pharmacology and Toxicology, Hebei Medical University, Shijiazhuang, China; [5]Yiling Pharmaceutical Company, Shijiazhuang, China; [6]Department of Clinical Pharmacy, Xingtai Ninth Hospital, Xingtai, China; [7]College of Chemical Engineering, Shijiazhuang University, Shijiazhuang, China; [8]Shijiazhuang Key Laboratory of Targeted Drugs Research and Efficacy Evaluation, Shijiazhuang, China; [9]Department of Pharmacy, Handan First Hospital, Handan, China; [10]Department of Psychiatry, The First Hospital of Hebei Medical University, Mental Health Institute of Hebei Medical University, Shijiazhuang, China

*For correspondence:
zhanghl@hebmu.edu.cn

### eLife assessment

This **important** study examined the mechanisms underlying reduced excitability of ventral tegmental area dopamine neurons in mice that underwent a chronic mild unpredictable stress treatment. The authors identify NALCN and TRPC6 channels as key mechanisms that regulate spontaneous firing of ventral tegmental area dopamine neurons and examined their roles in reduced firing in mice that underwent a chronic mild unpredictable stress treatment. The authors' conclusions on neurophysiological data are supported by multiple approaches and are **convincing**, although the relevance of the behavioral results to human depression remains unclear.

**Abstract** The slow-intrinsic-pacemaker dopaminergic (DA) neurons originating in the ventral tegmental area (VTA) are implicated in various mood- and emotion-related disorders, such as anxiety, fear, stress and depression. Abnormal activity of projection-specific VTA DA neurons is the key factor in the development of these disorders. Here, we describe the crucial role of the NALCN and TRPC6, non-selective cation channels in mediating the subthreshold inward depolarizing current and driving the firing of action potentials of VTA DA neurons in physiological conditions. Furthermore, we demonstrate that down-regulation of TRPC6 protein expression in the VTA DA neurons likely contributes to the reduced activity of projection-specific VTA DA neurons in chronic mild unpredictable stress (CMUS) depressive mice. In consistent with these, selective knockdown of TRPC6 channels in the VTA DA neurons conferred mice with depression-like behavior. This current

study suggests down-regulation of TRPC6 expression/function is involved in reduced VTA DA neuron firing and chronic stress-induced depression-like behavior of mice.

## Introduction

DA neurons in the VTA play a key role in mood, reward, and emotion-related behaviors (*Bromberg-Martin et al., 2010*). These DA neurons project, through the mesocorticolimbic dopaminergic system, to brain regions including the nucleus accumbens (NAc), medial prefrontal cortex (mPFC), and baso-lateral amygdala (BLA), and modulate these target neuronal circuits through their regulated activity (firing of action potentials) (*Lammel et al., 2014*; *Van den Heuvel and Pasterkamp, 2008*). As such, understandably, the altered functional activity of the VTA DA neurons is found to be the key determi-nant in abnormal behaviors related to the diseased conditions such as depression- or anxiety-related states (*Chaudhury et al., 2013*). Thus, the study of the mechanism underlying the firing activity of the VTA DA neurons is crucial for understanding the function of the DA circuitry and the pathogenesis of related mental diseases (*Chaudhury et al., 2013*; *Kabanova et al., 2015*; *Badrinarayan et al., 2012*; *Friedman et al., 2014*; *Krishnan et al., 2007*).

The VTA DA neurons are slow intrinsic pacemakers, which are further modulated by synaptic inputs from multiple excitatory and inhibitory projections (*Grace et al., 2007*; *Johnson and North, 1992*). Firing activity controls the release of DA, thus the function of the VTA DA neurons. The altered activity of the VTA DA neurons in a pathological state includes altered firing frequency and switch of the firing patterns (*Friedman et al., 2014*; *Zhong et al., 2018*).

The research on the mechanism for the spontaneous firing of action potential (AP) is the key to understand functional regulation of the VTA DA neurons. A variety of ion channels (e.g. voltage-gated Na⁺ channels, high-threshold-activated Ca²⁺ channels, Kv2, large conductance calcium-activated potassium channels, A- and M-type potassium channels) are reported to be involved in the regula-tion of the spontaneous firing activity of midbrain DA neurons (including the substantia nigra pars compacta, SNc and VTA) (*Bean, 2007*; *Guzman et al., 2009*; *Puopolo et al., 2007*; *Khaliq and Bean, 2010*; *Beckstead et al., 2004*; *Ford et al., 2007*; *Philippart et al., 2016*; *Dufour et al., 2014*; *Kimm et al., 2015*; *Li et al., 2017*; *Tarfa et al., 2017*). However, it is not completely clear how this spontaneous firing is initiated. AP fires when the resting membrane potential (RMP) is depolarized to the activation threshold for the voltage-dependent Na⁺ channels, which results in an opening of Na⁺ channels and the upstroke of AP (*Bean, 2007*). Therefore, a controlled depolarization of the RMP to the activation threshold of AP is the first key step in a cascade leading to the firing of AP. In the VTA DA neurons, the identity of ion channels contributing to this subthreshold depolarization has not been unambiguously established.

The hyperpolarization-activated cyclic-nucleotide-modulated (HCN1-4) channel gene family, one non-selective cation channel that would be activated at a hyperpolarized membrane potential (*Seutin et al., 2001*), represents the molecular correlates of the pacemaker current in some cells with sponta-neous firing of AP, such as If in cardiomyocytes and Ih in certain neurons including a subset of midbrain SNc DA neurons (*Seutin et al., 2001*). However, in the VTA DA neurons, results concerning the HCN channels to the spontaneous firing are controversial (*Khaliq and Bean, 2010*; *Seutin et al., 2001*; *Neuhoff et al., 2002*; *Krashia et al., 2017*).

Another non-selective cation channel, NALCN, unlike HCN, is a voltage-modulated channel (*Chua et al., 2020*), and is reported to be among the candidates for molecular correlates of the persistent background Na⁺ leak currents in the VTA DA neurons (*Lu et al., 2009*; *Philippart and Khaliq, 2018*). The NALCN-mediated background Na⁺ leak currents are found to be involved in the modulation of firing activity in a variety of neurons including DA neurons in the SNc (*Philippart and Khaliq, 2018*; *Shi et al., 2016*; *Lutas et al., 2016*; *Ford et al., 2018*; *Um et al., 2021*). However, it has not been reported if NALCN plays a role in the spontaneous firing of the VTA DA neurons, although the VTA DA neurons have larger Na⁺ leak currents than SNc DA neurons do (*Khaliq and Bean, 2010*).

TRP channels are a large class of non-selective cation channels, and are widely distributed in the peripheral and central nervous system. TRPC3, which has a high homology with TRPC6 (*Kim et al., 2007*), is found to be involved in the regulation of cardiac and SNc DA neuronal excitability, gener-ating depolarizing currents, triggering action potentials, and participating in cellular rhythmic firing

(*Um et al., 2021*; *Zhou et al., 2008*; *Hof et al., 2019*). The role of TRP channels, except TRPC4 (*Klipec et al., 2016*), in the regulation of VTA DA neuronal excitability has not been described.

In this study, we strove to find channel conductance which contribute to the subthreshold depolarization and thus the spontaneous firing of the VTA DA neurons. And furthermore, we investigated if these channels are the molecular mechanism for the altered function activity of the VTA DA neurons related to depression-like behaviors in mouse models of depression. We started by profiling the expression of non-selective cation channels (NSCCs) of the VTA DA neurons using the method of Patch-Seq from midbrain DA neurons projecting to different brain regions, and then focused on HCN, NALCN, TRPC6, and TRPV2 channels which are dominantly expressed in these DA neurons.

## Results

### Inflow of extracellular Na$^+$ contributes to the subthreshold depolarization of the VTA DA neurons

One key feature of the midbrain DA neurons is that these neurons are relatively depolarized in the resting condition; the resting membrane potential (RMP) is far from the K$^+$ equilibrium potential (~−55 mV *vs* ~−90 mV) (*Khaliq and Bean, 2010*), which enables the spontaneous firing. Thus, there must be persistent inward depolarizing conductance counterbalancing the hyperpolarizing K$^+$ conductance which otherwise would maintain the RMP at a hyperpolarized level. The main aim of this study is to identify the subthreshold depolarizing conductances which contribute to the spontaneous firing of the VTA DA neurons. While we initially performed our experiments using male mice, most of the electrophysiological experiments and some Western blot experiments were also repeated using female mice, results of which were reported in the following results. We first observed the firing frequency and the RMP of the VTA DA neurons using patch clamp recordings. The midbrain DA neurons, marked by dopamine transporter (DAT), are mainly located in the VTA and SNc (*Figure 1Ai*). We also performed single-cell PCR to identify the DA neurons with established markers for DA neurons (*Th*, *Slc6a3* (DAT), *Drd2* (Dopamine D2 Receptor), and *Kcnj6* (Girk2)). Although the majority of cells in the VTA area are DA neurons (~60%), there are still about 30% GABA neurons and a small number of glutamatergic neurons (*German and Manaye, 1993*; *Nair-Roberts et al., 2008*). Glutamatergic- and GABAergic neurons were also single-cell PCR typed by using markers of *Slc17a6* (Vglut2, Vesicular glutamate transporter 2) and *Gad1* (Glutamic acid decarboxylase 1), respectively (*Figure 1Aii*).

To focus on the intrinsic channel conductance for spontaneous firing, potential input modulation from fast-type transmitter transmission to the VTA DA neurons, namely activation of AMPA/kainite/NMDA receptor and GABA$_A$ receptor were blocked by receptor antagonists CNQX, APV, and picrotoxin (*Khaliq and Bean, 2010*), respectively, in the following experiments.

We first studied whether the inflow of extracellular Ca$^{2+}$ contributed to the initiation of spontaneous firing in the VTA DA neurons. It has been shown that, unlike the SNc DA neurons, the VTA DA neurons do not have subthreshold Ca$^{2+}$ oscillatory waves (*Khaliq and Bean, 2010*), suggesting that subthreshold Ca$^{2+}$ should not be an important component of subthreshold depolarizing conductance in the VTA DA neurons. In line with this finding, we found in our study, replacing Ca$^{2+}$ with Mg$^{2+}$ from the extracellular recording solution (ACSF, artificial cerebral spinal solution) did not reduce, but rather instead, increased the firing frequency of the VTA DA neurons in male and female mice (*Figure 1B*, from 1.77±0.14 Hz to 2.24±0.18 Hz, n=32, N=16, Wilcoxon matched-pairs signed rank test, W=526.0, p<0.0001). Consistent with this, the RMP of the VTA DA neurons was not hyperpolarized by replacing Ca$^{2+}$ with Mg$^{2+}$ from ACSF (*Figure 1C*, from −50.62±0.62 mV to −49.49±0.87 mV, n=25, N=15, Paired-sample t-test, t=2.368, df = 24, 95% CI: 0.1454–2.119, p=0.0263).

We then, in comparison with Ca$^{2+}$, observed the role of extracellular Na$^+$ in the RMP of the VTA DA neurons. For this, we followed the method of a previous study (*Khaliq and Bean, 2010*), by replacing Na$^+$ in ACSF with NMDG (N-methyl-d-glucamine); TTX (Tetrodotoxin) was also added to block the TTX-sensitive Na$^+$ currents, which is known to be the main contributor to the suprathreshold depolarization component of the action potential. Replacement of the extracellular Na$^+$ with NMDG resulted in a hyperpolarization of the RMP in male and female mice (from −48.11±0.99 mV to −59.93±1.51 mV, n=17, N=10, Paired-sample t-test, t=7.103, df = 16, 95% CI: −15.35 to −8.293, p<0.0001, *Figure 1D*), indicating a persistent inflow of Na$^+$ through a TTX-insensitive conductance which caused significant depolarization of the neurons, contributing significantly to the subthreshold depolarization of the

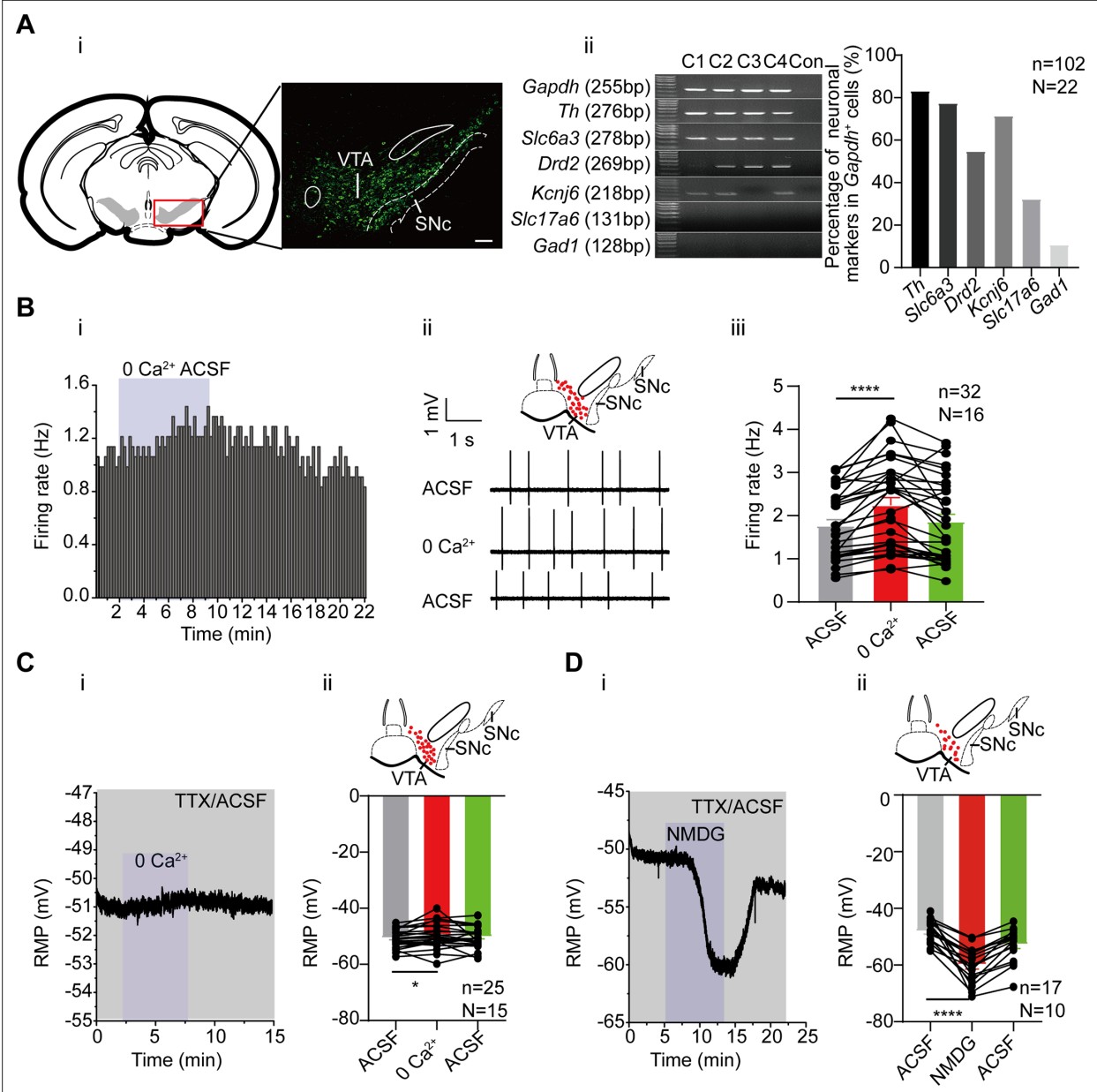

**Figure 1.** The effects of extracellular $Ca^{2+}$ and $Na^+$ on the firing activity of the male and female ventral tegmental area (VTA) dopaminergic (DA) neurons. (**A**) Identification of the VTA DA neurons. (**i**) Confocal images showing the anatomical distribution of DA neurons in the VTA and SNc; the DA neurons were identified to be dopamine transporter (DAT)-immunofluorescence positive (scale bar, 100 μm). (**ii**) left panel: single-cell PCR results; four cells (**C1–C4**) showed the presence of genes indicated. Right panel: Percentage of neuronal markers (DA neuron: *Th, Slc6a3, Drd2, Kcnj6*; Glu neuron: *Slc17a6* and GABA neuron: *Gad1*) positive neurons in the VTA neurons (*Gapdh* positive). n=102 cells, N=22. (**B**) The effect of replacing extracellular $Ca^{2+}$ with $Mg^{2+}$ on the firing frequency of the male and female VTA DA neurons. Example time-course (**i**) and traces (**ii**) of the spontaneous firing before and after replacement of extracellular $Ca^{2+}$ (2 mM) by an equimolar amount of $Mg^{2+}$. The inset on the top of (**ii**) shows a map of a coronal midbrain slice indicating the location of neurons that were recorded and subsequently identified as DA neurons which were *Slc6a3* positive with single cell-PCR (red dots). (**iii**) Summarized data for (**ii**) Firing of the VTA DA neurons was recorded using the loose cell-attached patch from a brain slice of the VTA (n=32, N=16). Wilcoxon matched-pairs signed rank test, W=526.0, ****p<0.0001. (**C**) The effect of replacing extracellular $Ca^{2+}$ with $Mg^{2+}$ on the resting membrane potential (RMP) of the male and female VTA DA neurons, in the presence of 1 μM TTX. Example time-course (**i**) and summarized data (**ii**) for the resting membrane potential before and after replacement of extracellular $Ca^{2+}$ (2 mM) by an equimolar amount of $Mg^{2+}$ (n=25, N=15). The inset on the top of (**ii**) shows a map of a coronal midbrain slice indicating the location of neurons that were recorded and subsequently identified as DA neurons which were *Slc6a3* positive with single cell-PCR (red dots). Paired-sample t-test, t=2.368, df = 24, 95% CI: 0.1454–2.119, *p=0.0263. (**D**) Replacement of external $Na^+$ by equimolar N-methyl-d-glucamine (NMDG) resulted in hyperpolarization of the resting membrane potential of the male and female VTA DA neurons, in the presence of 1 μM Tetrodotoxin (TTX). Example time-course (**i**) and summarized data (**ii**) for the resting membrane potential before and

*Figure 1 continued on next page*

*Figure 1 continued*

after replacement of extracellular Na$^+$ (151 mM) with an equimolar amount of NMDG (n=17, N=10). The inset on the top of (**ii**) shows a map of a coronal midbrain slice indicating the location of neurons that were recorded and subsequently identified as DA neurons which were *Slc6a3* positive with single cell-PCR (red dots). Paired-sample t-test, t=7.103, df=16, 95% CI: –15.35 to –8.293, ****p<0.0001.*p<0.05, ****p<0.0001. n is the number of neurons recorded and N is the number of mice used.

The online version of this article includes the following source data for figure 1:

**Source data 1.** Confocal image for the anatomical distribution of dopaminergic (DA) neurons in the ventral tegmental area (VTA) and SNc for *Figure 1Ai*.

**Source data 2.** Data and PDF file containing original gels for *Figure 1Aii*, indicating the relevant bands and groups.

**Source data 3.** Original files for gels of single-cell PCR are displayed in *Figure 1Aii*.

**Source data 4.** Data of the effect of extracellular Ca$^{2+}$ on the firing frequency of the VTA DA neurons for *Figure 1B*.

**Source data 5.** Data of the effect of extracellular Ca$^{2+}$ on the resting membrane potential (RMP) of the ventral tegmental area dopaminergic (VTA DA) neurons for *Figure 1C*.

**Source data 6.** Data of the effect of Na$^+$ in resting membrane potential (RMP) of the ventral tegmental area dopaminergic (VTA DA) neurons for *Figure 1D*.

VTA DA neurons. Taken together, the above results indicate persistent Na$^+$ influx contributes to the subthreshold depolarization of the VTA DA neurons.

## Potential candidates of channel conductance contributing to subthreshold depolarization of the VTA DA neurons

The channel conductance mediating the above-described persistent Na$^+$ influx should come from a cation channel(s) permeating to Na$^+$. We thus investigated the presence and expression level of non-selective cation channels (NSCCs) in the VTA DA neurons using a combination of patch clamp and single-cell RNA-seq (Patch-seq) technology (*Cadwell et al., 2016*). In this part of the study, we considered the heterogeneity and projection-specificity of the VTA DA neurons, i.e., VTA DA neurons projecting to different brain regions have different electrophysiological characteristics (*Lammel et al., 2008*). Using retrograde labeling techniques by retrobeads, 60 male VTA neurons projecting to five different brain regions (medial prefrontal cortex, mPFC; basal lateral amygdala, BLA; nucleus accumbens core, NAc c; nucleus accumbens lateral shell, NAc ls; nucleus accumbens medial shell, NAc ms) (*Figure 2—figure supplement 1*) were collected for RNA-seq, using patch-clamp electrodes. Consistent with previous studies (*Lammel et al., 2008*), the VTA DA neurons projecting to above different brain regions were anatomically congregated into subregions of the VTA (*Figure 2—figure supplement 2*). The high expression of biomarkers (Tyrosine hydroxylase, *Th*; Dopamine decarboxylase, *Ddc* and Dopamine transporter, *Slc6a3*) in the 45 cells of 60 cells (*Figure 2A*) indicates that these are DA neurons. In addition, fewer than half of sequenced DA neurons (predominantly those projecting to NAc ms, BLA, and mPFC) have biomarkers (Vesicular glutamate transporter 1–3, *Slc17a6-8*) of glutamatergic neurons, predicting a possible coexistence of neurotransmitters (DA and glutamate) (*Kawano et al., 2006*). In contrast, fewer GABAergic neuronal markers (Glutamic acid decarboxylase, *Gad1/2* and Vesicular GABA transporter, *Slc32a1*) co-expressed with the DA neurons, which is consistent with previous studies that VTA DA neurons which co-release GABA, contain low and even undetectable levels of GAD1, GAD2 and VGAT mRNA or protein (*Patel et al., 2024*; *Tritsch et al., 2016*). It needs to be noted that some neurons in the VTA only expressed markers for glutamatergic or GABAergic neurons, these neurons were excluded from further analysis as we focused our study on the DA neurons.

Expression levels of NSCCs in the VTA DA neurons classified by projection specificity were analyzed and shown in *Figure 2B*. Different colored strips on the top of the figure corresponded to the projection specificity coded by the same colors in *Figure 2A*. Multiple transient receptor potential channels (TRP) were present in the VTA DA neurons, with prominent expression level of *Trpc6* and *Trpv2* (*Figure 2Bi*). Other prominently expressed NSCCs included *Hcn2*, *Hcn3*, *Nalcn*, and *Panx1* (Pannexin 1) (*Figure 2Bii*). Summarized averaged relative expression levels for the eight most richly expressed NSCCs are shown in *Figure 2Biii*. Furthermore, projection-specific expression of these dominantly expressed NSCCs was analyzed and shown in *Figure 2C*; interestingly only *Trpc6* seemed expressed in a projection-specific manner, namely, the VTA DA neurons projecting to the NAc have higher *Trpc6*

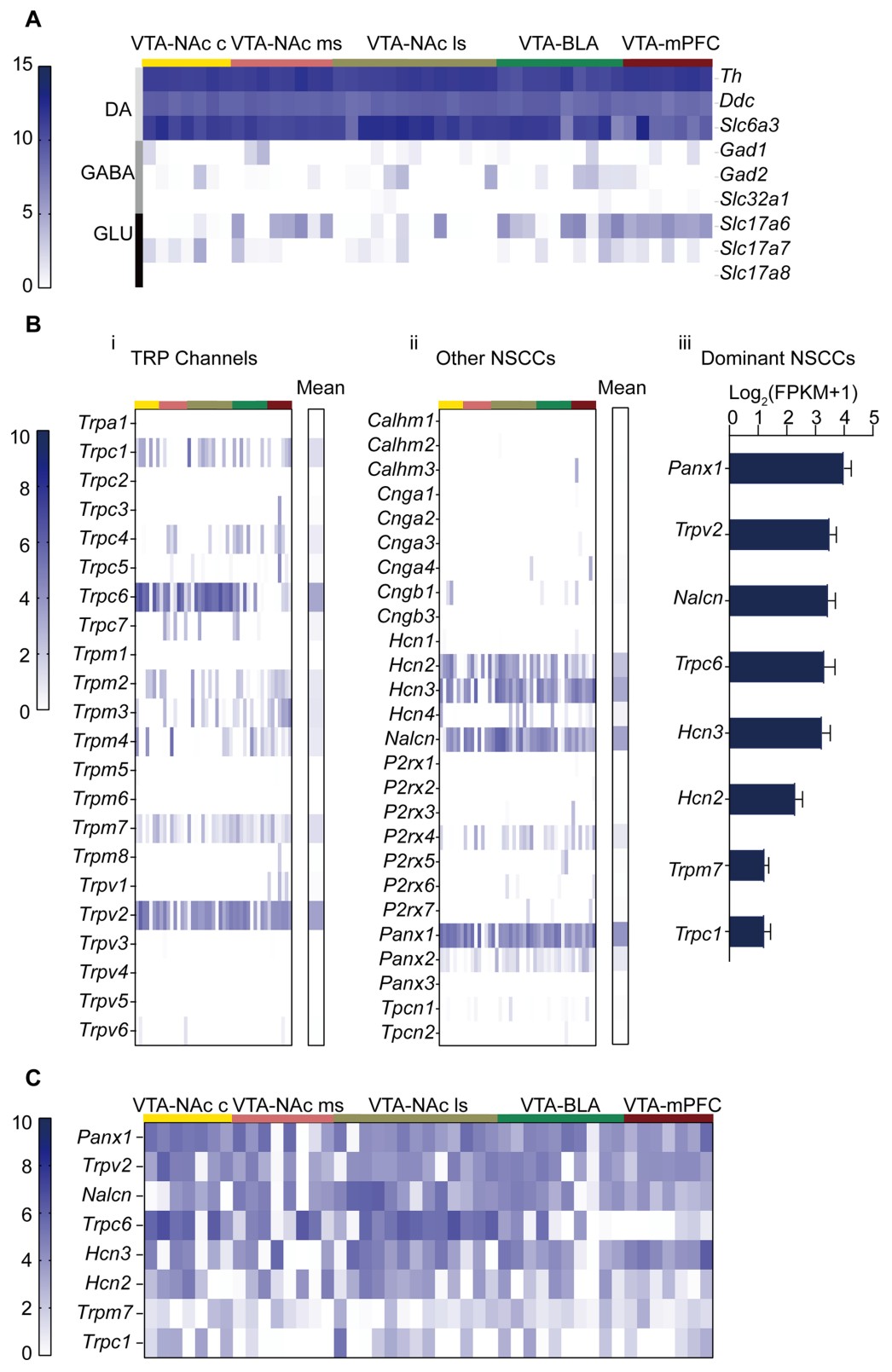

**Figure 2.** Gene expression profile of non-selective cation channels (NSCCs) in the male ventral tegmental area dopaminergic (VTA DA) neurons. Single-cell RNA-seq was performed on the VTA neurons projecting to five different brain regions of male mice (NAc c, NAc ms, NAc ls, BLA, mPFC). (**A**). Gene expression profile of markers for neuron subtypes from 45 VTA neurons; the neuron subtypes included: dopaminergic (*Th*, *Ddc*, *Slc6a3*),

*Figure 2 continued on next page*

*Figure 2 continued*

GABAergic (*Gad1*, *Gad2*, *Slc32a1*), and glutamatergic (*Slc17a6*, *Slc17a7*, *Slc17a8*), which was arranged in different rows indicated in the right labels and in the left-colored vertical lines; projection targets of these neurons were indicated at the top by the colored lines and labels. Relative expression levels of these genes were indicated by the dark-blue color intensity which was transformed from the log2 values of the number of transcripts per million (FPKM) plus 1. (**B**). Relative gene expression levels of transient receptor potential (TRP) channels (**i**) and other NSCCs (**ii**) in 45 individual VTA DA neurons and the population average (mean, right columns). Each column of the individual neurons in (**i**) and (**ii**) corresponded to the columns in A. (**iii**) Bar graph of the mean log2 (FPKM +1) for the top eight NSCCs in descending order. Error bars indicate SEM. (**C**) Gene expression profile of aforementioned top eight NSCCs from 45 VTA neurons.

The online version of this article includes the following source data and figure supplement(s) for figure 2:

**Source data 1.** Gene expression data for *Figure 2*.

**Figure supplement 1.** The targeted regions for ventral tegmental area dopaminergic (VTA DA) projection.

**Figure supplement 1—source data 1.** PDF file containing confocal images for *Figure 2—figure supplement 1*.

**Figure supplement 2.** The subregions of projection-specific ventral tegmental area dopaminergic (VTA DA) neurons.

**Figure supplement 2—source data 1.** PDF file containing confocal images for *Figure 2—figure supplement 2*.

**Figure supplement 3.** Blocking HCN channels does not affect the spontaneous firing of the ventral tegmental area dopaminergic (VTA DA) neurons in adult male mice.

**Figure supplement 3—source data 1.** Data of the effects of CsCl on the sag membrane potential of ventral tegmental area dopaminergic (VTA DA) neurons for *Figure 2—figure supplement 3A*.

**Figure supplement 3—source data 2.** Data of the effects of ZD7288 on the sag membrane potential of ventral tegmental area dopaminergic (VTA DA) neurons for *Figure 2-figure supplement 3B*.

**Figure supplement 3—source data 3.** Data of the effects of CsCl on the excitability of ventral tegmental area dopaminergic (VTA DA) neurons for *Figure 2—figure supplement 3C*.

**Figure supplement 3—source data 4.** Data of the effects of ZD7288 on the excitability of ventral tegmental area dopaminergic (VTA DA) neurons for *Figure 2—figure supplement 3D*.

**Figure supplement 4.** HCN channel blocker does not affect the spontaneous firing of the adult male ventral tegmental area dopaminergic (VTA DA) neurons projecting to NAc lateral shell but blocks the spontaneous firing of the VTA DA neurons bathed in a low extracellular $K^+$ which hyperpolarized the resting membrane potential.

**Figure supplement 4—source data 1.** Data of the effects of CsCl on the excitability of ventral tegmental area (VTA) NAc ls-projecting dopaminergic (DA) neurons for *Figure 2—figure supplement 4A*.

**Figure supplement 4—source data 2.** Data of the effects of ZD7288 on the excitability of VTA NAc ls-projecting dopaminergic (DA) neurons for *Figure 2—figure supplement 4B*.

**Figure supplement 4—source data 3.** Data of the effect of low extracellular $K^+$ on the resting membrane potential (RMP) of ventral tegmental area dopaminergic (VTA DA) neurons for *Figure 2—figure supplement 4C*.

**Figure supplement 4—source data 4.** Data of the effect of ZD7288 on the excitability of ventral tegmental area dopaminergic (VTA DA) neurons in low $[K^+]_e$ condition for *Figure 2—figure supplement 4D*.

**Figure supplement 4—source data 5.** Data of the effects of ZD7288 on the excitability of juvenile male ventral tegmental area dopaminergic (VTA DA) neurons for *Figure 2—figure supplement 4E*.

expression than the VTA DA neurons projecting to the BLA and the mPFC (*Figure 2C*). In our subsequent study, we focused our study on HCN, NALCN, TRPC6, and TRPV2, one for they are the prominent expression channels, and two for some of these channels have been indicated in the modulation of excitability of different neuron types (*Seutin et al., 2001*; *Neuhoff et al., 2002*; *Philippart and Khaliq, 2018*; *Shi et al., 2016*; *Lutas et al., 2016*; *Ford et al., 2018*; *Um et al., 2021*; *Lu et al., 2007*). Panx1 was investigated in a separate study since it normally composes the semi channels which are different from these more conventional ion channels.

## HCN does not contribute to the spontaneous firing of the VTA DA neurons

We mainly used two pharmacological tools as HCN channel blockers, CsCl and ZD7288 (*Khaliq and Bean, 2010*; *Neuhoff et al., 2002*) to explore the role of HCN channels in the spontaneous firing of

the male VTA DA neurons. Efficient blocking effect of these blockers on the HCN channel was first verified. For this, the sag potential produced by a hyperpolarizing current (–100 pA) injection was used to assess HCN activity (*Neuhoff et al., 2002*). Both CsCl (3 mM) and ZD7288 (60 µM) effectively reduced the sag potential of VTA DA neurons in male mice (from 16.43±2.96 mV to 3.84±1.92 mV, n=4, N=4, Paired-sample t-test, t=4.246, df = 3, 95% CI: –22.03 to –3.155, p=0.0239; from 19.38±3.14 mV to 4.18±2.60 mV, n=4, N=4, Paired-sample t-test, t=5.017, df = 3, 95% CI: –24.83 to –5.555, p=0.0152, respectively) (*Figure 2—figure supplement 3A and B*).

However, these same HCN blockers did not inhibit the spontaneous firing of the VTA DA neurons in male mice; CsCl slightly increased (1.96±0.19 Hz to 2.14±0.34 Hz, n=7, N=4, Paired-sample t-test, t=0.7171, df = 6, 95% CI: –0.4325–0.7911, p=0.5003) whereas ZD7288 slightly decreased the firing frequency (1.65±0.25 Hz to 1.50±0.24 Hz, n=13, N=5, Paired-sample t-test, t=1.349, df = 12, 95% CI: –0.4033–0.09486, p=0.2022), but none of these effects was statistically significant (p>0.05) (*Figure 2—figure supplement 3C and D*). It has been reported that VTA DA neurons projecting to the NAc lateral shell have more pronounced HCN/Ih currents than other VTA DA neurons (*Lammel et al., 2008*). However, even the male VTA DA neurons projecting to the NAc lateral shell (NAc ls, retrogradely labeled by retrobeads), no effect of CsCl or ZD7288 on the spontaneous firing of these neurons were observed (from 2.04±0.24 Hz to 1.93±0.22 Hz, n=5, N=4, Paired-sample t-test, t=0.4798, df=4, 95% CI: –0.7601–0.5361, p=0.6564 by CsCl, and from 1.71±0.29 Hz to 1.65±0.33 Hz, n=5, N=5, Paired-sample t-test, t=0.4421, df=4, 95% CI: –0.4368–0.3168, p=0.6813 by ZD7288) (*Figure 2—figure supplement 4A and B*).

HCN is activated by membrane hyperpolarization, with an activation threshold around –70~–90 mV (HCN2, HCN3) (*Kusch et al., 2010*; *Mistrík et al., 2005*). At a depolarized RMP like these in the VTA DA neurons (–51.28±1.85 mV, n=8, N=8; e.g. –54.50 mV, *Figure 2—figure supplement 3Ai*, and –53.50 mV, *Figure 2—figure supplement 3Bi*), HCN are most likely not activated thus would not participate in the generation of spontaneous firing. To further prove this, we tested if a more hyperpolarized RMP would involve HCN in the spontaneous firing of the VTA DA neurons. For this, we first lowered the $K^+$ concentration in the extracellular ACSF from 3 mM to 1.5 mM to increase the gradient between inside and outside cellular $K^+$; this maneuver indeed hyperpolarized the cell membrane from –49.45±0.93 mV to –62.95±2.56 mV (n=7, N=7, Paired-sample t-test, t=6.373, df = 6, 95% CI: –18.69 to –8.318, p=0.0007) (*Figure 2—figure supplement 4C*). Interestingly, under 1.5 mM extracellular $K^+$, the spontaneous firing of the male VTA DA neurons was almost totally blocked by ZD7288 (*Figure 2—figure supplement 4D*), although an initial enhancement was seen in some neurons (*Figure 2—figure supplement 4Di*).

We tested another possible mechanism for HCN's failure to participate in the generation of VTA DA firing. It has been reported that in the midbrain SNc DA neurons, HCN is involved in the spontaneous firing in young but not in adult mice, due to a more hyperpolarization-shifted activation threshold of HCN in SNc DA neurons in adult mice. We tested if this could also be the case in the VTA DA neurons. Indeed, as shown in *Figure 2—figure supplement 4E*, in contrast to what we saw in adult male mice, the spontaneous firing frequency of the VTA DA neurons in young male mice (less than 15 d postnatal) was significantly reduced by ZD7288 (from 1.86±0.29 Hz to 0.98±0.24 Hz, n=9, N=5, Paired-sample t-test, t=3.345, df=8, 95% CI: –1.487 to –0.2734, p=0.0101).

Taken together, the above results suggest HCN is not involved in the spontaneous firing of the VTA DA neurons in adult male mice, due to a depolarized RMP and a hyperpolarization-shifted activation property.

## NALCN contributes to subthreshold depolarization and spontaneous firing of the VTA DA neurons

The $Na^+$ currents produced by NALCN have been suggested to be an important component of background $Na^+$ currents in multiple central neurons including DA neurons in the substantia nigra (*Philippart and Khaliq, 2018*). We next investigated if NALCN also plays a role in setting the RMP and in the spontaneous firing of the VTA DA neurons. Consistent with the above RNA-seq results, both immunofluorescence (*Figure 3A*) and single-cell PCR (*Figure 3B*) results confirmed broad expression of NALCN in the VTA DA neurons.

GdCl₃ (100 µM), a non-selective NALCN blocker (*Lutas et al., 2016*; *Ford et al., 2018*), hyperpolarized the RMP of the male VTA DA neurons from –48.88±2.41 mV to –68.88±1.28 mV (n=7,

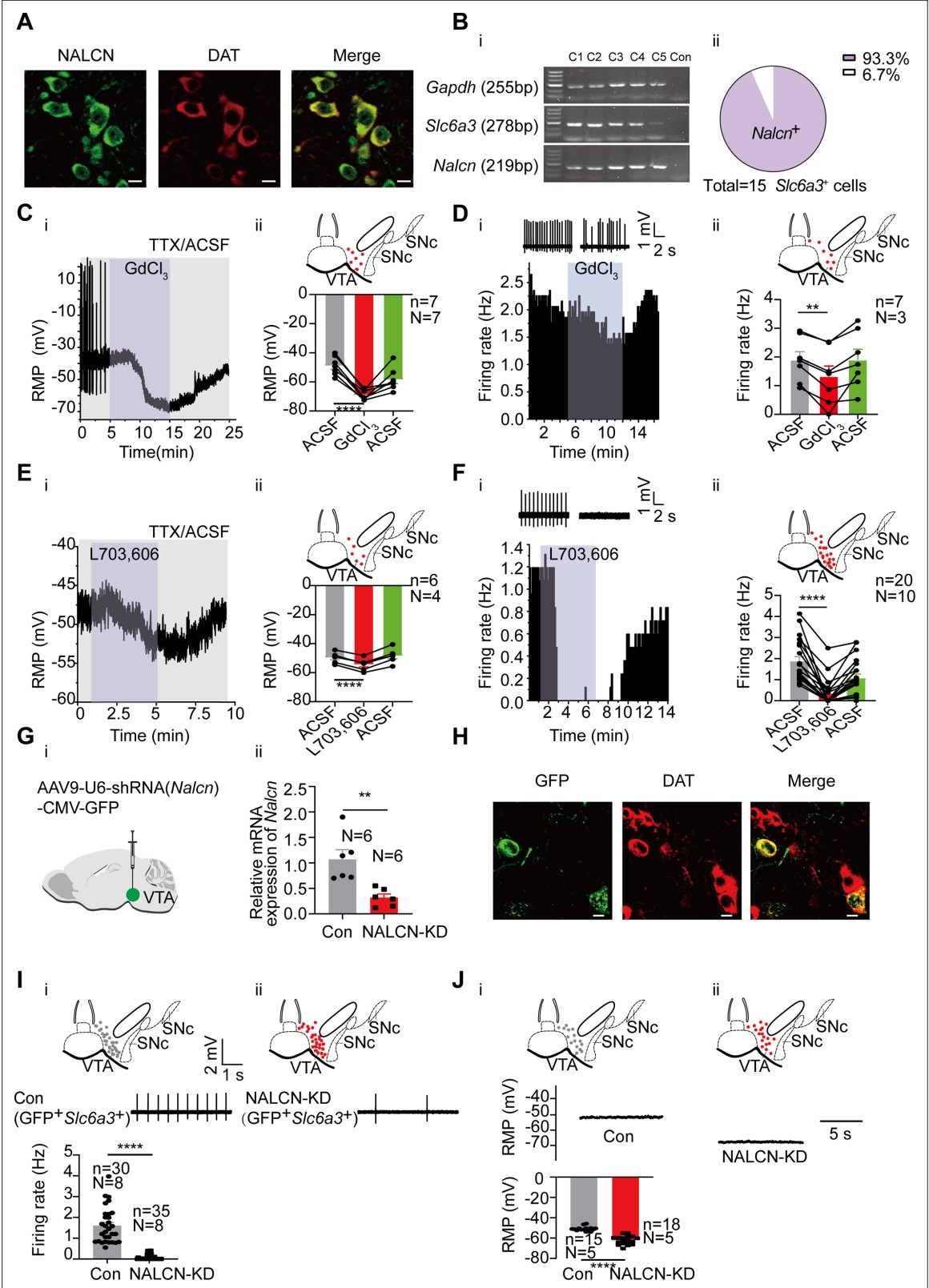

**Figure 3.** NALCN contributes to subthreshold depolarization and spontaneous firing of the ventral tegmental area dopaminergic (VTA DA) neurons. (**A**) Confocal images showing co-expression of NALCN (green) and DAT (red) representing DA neurons (scale bar, 10 μm). (**B**) (**i**) Single-cell PCR from the VTA DA neurons (**C1–C5**) with the expression of *Nalcn*. (**ii**) Percentage of *Nalcn* positive neurons from 15 DA neurons (with expression of *Slc6a3*). (**C**) Example time-course (**i**) and summarized data (**ii**) showed that NALCN channel blocker (GdCl$_3$) significantly hyperpolarized the resting membrane

*Figure 3 continued on next page*

*Figure 3 continued*

potential (RMP) (n=7, N=7) of male VTA DA neurons. The inset on the top of Cii shows a map of a coronal midbrain slice indicating the location of neurons that were recorded and subsequently identified as DA neurons which were *Slc6a3* positive with single cell-PCR (red dots). Paired-sample t-test, t=11.38, df=6, 95% CI: –24.30 to –15.70, ****p<0.0001. (**D**) Example time-course and traces (**i**) and summarized data (**ii**) of the effect of GdCl₃ on spontaneous firing frequency in male VTA DA neurons (n=7, N=3). The inset on the top of Dii shows a map of a coronal midbrain slice indicating the location of neurons that were recorded and subsequently identified as DA neurons which were *Slc6a3* positive with single cell-PCR (red dots). Paired-sample t-test, t=5.711, df=6, 95% CI: –0.8183 to –0.3274, **p=0.0012. (**E**) Example time-course (**i**) and summarized data (**ii**) showed that NALCN channel blocker (L703,606) significantly hyperpolarized the resting membrane potential (RMP) (n=6, N=4) of VTA DA neurons in male mice. The inset on the top of **Eii** shows a map of a coronal midbrain slice indicating the location of neurons that were recorded and subsequently identified as DA neurons which were *Slc6a3* positive with single cell-PCR (red dots). Paired-sample t-test, t=14.95, df=5, 95% CI: –5.498 to –3.885, ****p<0.0001. (**F**) Example time-course and traces (**i**) and summarized data (**ii**) of the effect of L703,606 on spontaneous firing frequency in DA neurons of both sexes (n=20, N=10). The inset on the top of **Fii** shows a map of a coronal midbrain slice indicating the location of neurons that were recorded and subsequently identified as DA neurons which were *Slc6a3* positive with single cell-PCR (red dots). Wilcoxon matched-pairs signed rank test, W=–210.0, ****p<0.0001. (**G**). shRNA against *Nalcn* carried by AAV virus (**i**) was injected into the VTA of mice, the mRNA level in the shRNA-*Nalcn* transfected VTA (NALCN-KD, N=6), and the scramble shRNA transfected VTA (Control, N=6) was analyzed using qPCR (**ii**). Two-sample t-test, t=3.714, df=10, 95% CI: –1.200 to –0.3000, **p=0.0040. (**H**) Confocal images showing the expression of AAV9-*Nalcn*-shRNA-GFP (green) in the VTA DA neurons (DAT, red) (scale bar, 10 μm). (**I**). Loose cell-attached current clamp recordings of the spontaneous firing of the male and female VTA DA neurons transfected with either *Nalcn*-shRNA (**ii**, n=35, N=8) or scramble-shRNA (Con, **i**, n=30, N=8) (both GFP and *Slc6a3* positive). Examples of firing traces in the middle of **i** and **ii** and summarized data in the bottom of **i** were shown. The inset on the top of (**i** and **ii**) shows a map of a coronal midbrain slice indicating the location of GFP⁺ neurons that were recorded and subsequently identified as DA neurons which were *Slc6a3* positive with single cell-PCR (**i**: Con, gray dots; **ii**: NALCN-KD, red dots). Mann-Whitney U test, U=0, **** p<0.0001. (**J**). Whole-cell current clamp recordings of the resting membrane potential (RMP) of the male VTA DA neurons transfected with either *Nalcn*-shRNA (NALCN-KD, **ii**, n=18, N=5) or scramble-shRNA (Con, **i**, n=15, N=5) (both GFP and *Slc6a3* positive). Examples of RMP traces in the middle of **i** and **ii** and summarized data in the bottom of i were shown. The inset on the top of **i** and **ii** shows a map of a coronal midbrain slice indicating the location of GFP⁺ neurons that were recorded and subsequently identified as DA neurons which were *Slc6a3* positive with single cell-PCR (**i**: Con, gray dots; **ii**: NALCN-KD, red dots). Mann-Whitney U test, U=0, ****p<0.0001. *p<0.05, **p<0.01, ****p<0.0001. n is the number of neurons recorded and N is the number of mice used.

The online version of this article includes the following source data for figure 3:

**Source data 1.** PDF file containing confocal images for *Figure 3A and H*.

**Source data 2.** PDF file containing original gels of single-cell PCR for *Figure 3Bi*, indicating the relevant bands and groups.

**Source data 3.** Original files for gels of single-cell PCR are displayed in *Figure 3Bi*.

**Source data 4.** Data on the effect of GdCl₃ on the resting membrane potential (RMP) of ventral tegmental area dopaminergic (VTA DA) neurons is displayed in *Figure 3C*.

**Source data 5.** Data on the effect of GdCl₃ on the firing frequency of ventral tegmental area dopaminergic (VTA DA) neurons is displayed in *Figure 3D*.

**Source data 6.** Data on the effect of L703,606 on the resting membrane potential (RMP) of ventral tegmental area dopaminergic (VTA DA) neurons is displayed in *Figure 3E*.

**Source data 7.** Data on the effect of L703,606 on the firing frequency of ventral tegmental area dopaminergic (VTA DA) neurons is displayed in *Figure 3F*.

**Source data 8.** Data of the expression level of *Nalcn* RNA in the shRNA-*Nalcn* transfected ventral tegmental area (VTA) and the scramble shRNA transfected VTA for *Figure 3G*.

**Source data 9.** Data of excitability of ventral tegmental area dopaminergic (VTA DA) neurons transfected with either *Nalcn*-shRNA or scramble-shRNA for *Figure 3I*.

**Source data 10.** Data of resting membrane potential (RMP) of ventral tegmental area dopaminergic (VTA DA) neurons transfected with either *Nalcn*-shRNA or scramble-shRNA for *Figure 3J*.

N=7, Paired-sample t-test, t=11.38, df=6, 95% CI: –24.30 to –15.70, p<0.0001) (*Figure 3C*), reduced the spontaneous firing frequency of the male VTA DA neurons from 1.89±0.29 Hz to 1.32±0.37 Hz (n=7, N=3, Paired-sample t-test, t=5.711, df=6, 95% CI: –0.8183 to –0.3274, p=0.0012) (*Figure 3D*). L703,606 (10 μM), a selective NALCN blocker (*Um et al., 2021*), also hyperpolarized the RMP of the male VTA DA neurons from –49.61±1.48 mV to –54.30±1.67 mV (n=6, N=4, Paired-sample t-test, t=14.95, df=5, 95% CI: –5.498 to –3.885, p<0.0001) (*Figure 3E*), reduced the spontaneous firing frequency of the male and female VTA DA neurons from 1.87±0.23 Hz to 0.34±0.14 Hz (n=20, N=10, Wilcoxon matched-pairs signed rank test, W=–210.0, p<0.0001) (*Figure 3F*).

To observe a more specific effect on NALCN, a shRNA against *Nalcn* was used to knockdown NALCN. For this, AAV9 viral construct (AAV9-U6-shRNA(*Nalcn*)-CMV-GFP) was injected into the VTA of a mouse; a similar AAV viral construct containing scramble-shRNA of nonsense sequences was utilized

as controls. The qPCR results show sufficient knockdown of *Nalcn* mRNA (NALCN-KD: 0.32±0.07, N=6; Con: 1.07±0.19, N=6, Two-sample t-test, t=3.714, df=10, 95% CI: –1.200 to –0.3000, p=0.0040, *Figure 3G*) in the VTA tissue. Immunofluorescence results (*Figure 3H*) show efficient infection of the VTA DA neurons (GFP expression in the DAT-positive neurons) by the virus. Knockdown of NALCN in the VTA DA neurons by shRNA almost completely silenced the firing of the male and female VTA DA neurons (0.08±0.02 Hz *vs* 1.60±0.15 Hz in NALCN-KD (n=35, N=8) and Con (n=30, N=8) infected VTA DA neurons, respectively, Mann-Whitney U test, U=0, p<0.0001, *Figure 3I*). Furthermore, the RMP of the male VTA DA neurons was significantly hyperpolarized (–62.13±1.00 mV *vs* –50.86±0.57 mV in *Nalcn*-shRNA and scramble-shRNA infected VTA DA neurons, respectively, n=18 and 15, N=5 and 5, Mann-Whitney U test, U=0, p<0.0001, *Figure 3J*).

Taken together, the above results suggest that NALCN is a major contributor to the subthreshold depolarization and contributes significantly to the generation of spontaneous firing in the VTA DA neurons.

## TRPC6 contributes to subthreshold depolarization and spontaneous firing of the VTA DA neurons

Above RNA-seq results suggest a broad and strong expression of TRPC6 and TRPV2 channels in the VTA DA neurons. This part of the experiments was focused on the role of these two TRP channels in subthreshold depolarization and spontaneous firing of the VTA DA neurons.

Consistent with the RNA-seq results, single-cell PCR experiments confirmed high proportion expression of *Trpc6* and *Trpv2* in the VTA DA neurons (*Figure 4A*); among the 28 VTA DA neurons (*Slc6a3*⁺), 20 neurons expressed *Trpv2* (71.4%), and 18 neurons (64.3%) expressed *Trpc6*. TRPC3, which is highly homologous with TRPC6 (*Clapham et al., 2001*) and has been reported to be involved in the firing activity of the SNc DA neurons (*Um et al., 2021*), was neither detected in RNA-seq study (*Figure 2B*), nor in the single-cell PCR experiment (*Figure 4A*). These results reciprocally confirmed the reliability of both the RNA-seq and single cell PCR results.

General roles of TRP channels especially TRPC6 in subthreshold depolarization and spontaneous firing of the VTA DA neurons were first assessed using a non-selective broad-spectrum TRP channel blocker, 2-aminoethoxydiphenylborane (2-APB; 100 µM) and a selective TRPC6 channel blocker, larixyl acetate (*Urban et al., 2016*) (LA;10 µM). Both blockers significantly reduced the firing frequency of the VTA DA neurons (2-APB for male mice: from 1.32±0.27 Hz to 0.06±0.02 Hz, n=12, N=5, Wilcoxon matched-pairs signed rank test, W=–78.0, p=0.0005; LA for male and female mice: from 1.59±0.26 Hz to 0.55±0.15 Hz, n=14, N=8, Wilcoxon matched-pairs signed rank test, W=–105.0, p=0.0001; *Figure 4B and C*). Accordingly, both blockers also significantly hyperpolarized the RMP of the male and female VTA DA neurons (2-APB: from –49.97±1.29 mV to –59.27±1.22 mV, n=10, N=9, Paired-sample t-test, t=8.082, df=9, 95% CI: –11.90 to –6.693, p<0.0001; LA: from –53.41±0.81 mV to –57.82±1.13 mV, n=9, N=7, Paired-sample t-test, t=6.431, df=8, 95% CI: –5.993 to –2.829, p=0.0002; *Figure 4D and E*).

We then tested a more TRPV channel-selective blocker ruthenium red (RR), to study the possible role of TRPV2 (and other TRPV) in the spontaneous firing of the VTA DA neurons. RR (60 µM), on average statistically, did not affect the spontaneous firing of the male VTA DA neurons (from 2.30±0.31 Hz to 2.18±0.31 Hz, n=9, N=6, Paired-sample t-test, t=0.4486, df=8, 95% CI: –0.7621–0.5139, p=0.6656) (*Figure 4—figure supplement 1*). However, it needs to be noted that among the nine VTA DA neurons, we tested, both increase (*Figure 4—figure supplement 1B*) and decrease (*Figure 4—figure supplement 1C*) of firing frequency by RR was seen, thus with opposite changes of firing frequency in different populations of neurons (*Figure 4—figure supplement 1A*), no overall effect of RR was obtained from the current analysis.

We then focused our next study on TRPC6. Consistent with both RNA-seq (*Figure 2B*) and single-cell PCR (*Figure 4A*) results, the mRNA level of *Trpc6* in mPFC-projecting VTA single DA neuron is significantly lower than that in NAc c-projecting VTA single DA neuron (0.56±0.12, n=25, N=8 for mPFC-projecting TH-positive cells *vs* 1.09±0.10, n=28, N=7 for NAc c-projecting TH-positive cells, Mann-Whitney U test, U=173, p=0.0013, *Figure 5A*) The immunofluorescence results also demonstrated strong expression of TRPC6 protein in the VTA DA neurons (*Figure 5B*, TH-positive). An AAV viral construct with shRNA against *Trpc6* (AAV9-U6-shRNA(*Trpc6*)-CMV-GFP) was injected into the VTA of mice, to knockdown TRPC6; efficiency of this knockdown was assessed by using qPCR measuring

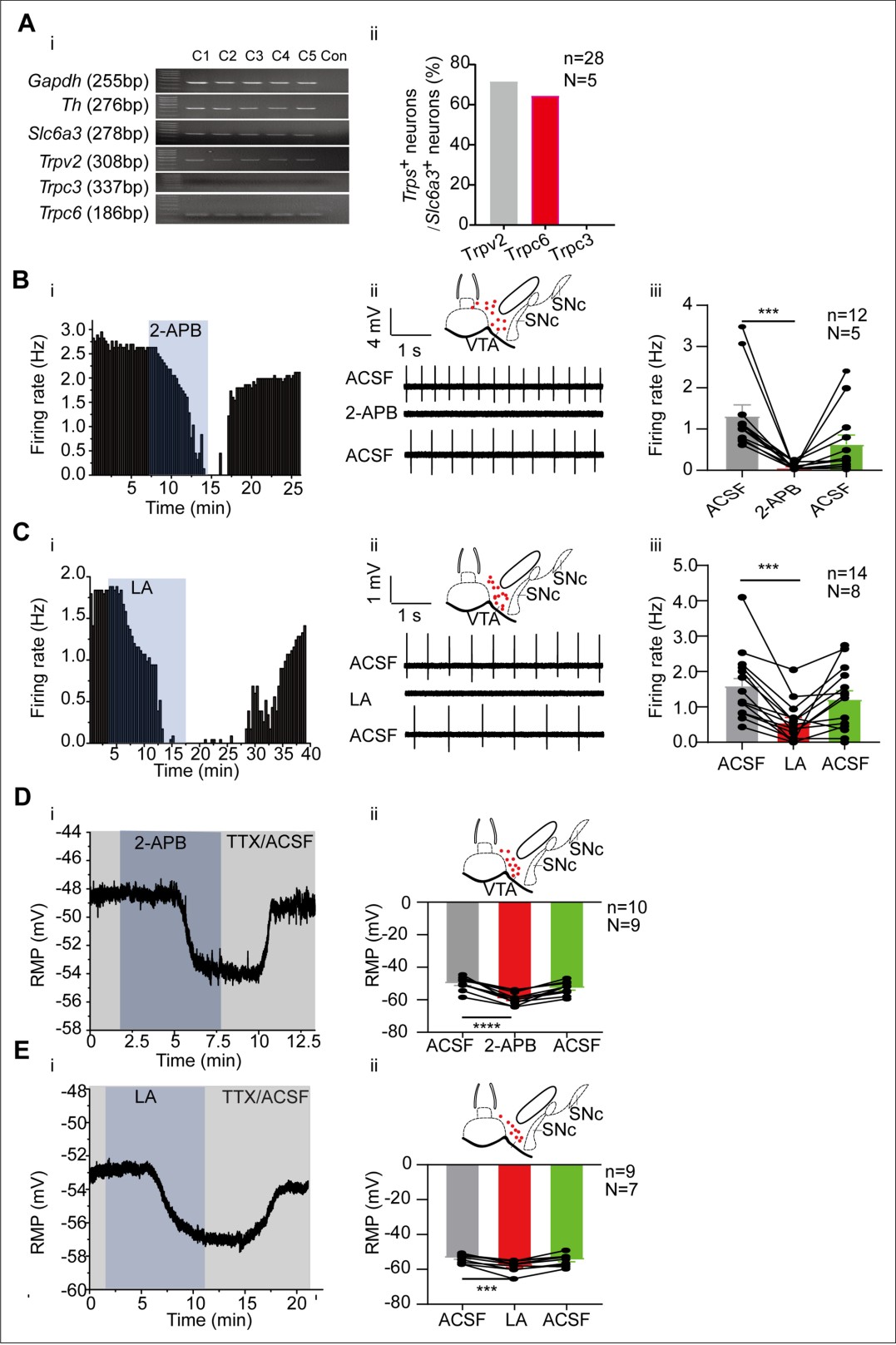

**Figure 4.** Transient receptor potential (TRP) channels especially TRPC6 contribute to subthreshold depolarization and spontaneous firing of the ventral tegmental area dopaminergic (VTA DA) neurons. (**A**) The expression of TRP channels in VTA DA neurons. (**i**) Single-cell PCR from 5 VTA cells (C1–C5). (**ii**) Percentage of TRP channels (*Trpc3*, *Trpc6* and *Trpv2*) positive neurons in the VTA DA neurons (*Slc6a3* positive). (**B and C**) Example time-course

*Figure 4 continued on next page*

*Figure 4 continued*

(i), traces (ii) and the summarized data (iii) for the effect of a nonselective cation channel blocker 2-APB (100 µM, n=12, N=5 for male mice) (B) and a potent TRPC6 inhibitor LA (10 µM, n=14, N=8 for male and female mice) (C) on the spontaneous firing frequency in the VTA DA neurons. The inset on the top of **Bii** and **Cii** shows a map of a coronal midbrain slice indicating the location of neurons that were recorded and subsequently identified as DA neurons which were *Slc6a3* positive with single cell-PCR (red dots). Wilcoxon matched-pairs signed rank test, **B**: W=−78.00, ***p=0.0005. **C**: W=−105.0, ***p=0.0001. (**D and E**) Example time-course (i) and the summarized data (ii) for the effect of a nonselective cation channel blocker 2-APB (n=10, N=9) (**D**) and a potent TRPC6 inhibitor LA (n=9, N=7) (**E**) on the resting membrane potential (RMP) of the male and female VTA DA neurons. The inset on the top of Dii and Eii shows a map of a coronal midbrain slice indicating the location of neurons that were recorded and subsequently identified as DA neurons which were *Slc6a3* positive with single cell-PCR (red dots). Paired-sample t-test, (**D**) t=8.082, df=9, 95% CI: −11.90 to −6.693, ****p<0.0001, (**E**) t=6.431, df = 8, 95% CI: −5.993 to −2.829, ***p=0.0002. ***p<0.001, ****p<0.0001. n is the number of neurons recorded and N is the number of mice used.

The online version of this article includes the following source data and figure supplement(s) for figure 4:

**Source data 1.** Data and a PDF file containing original gels of single-cell PCR for *Figure 4A*, indicating the relevant bands and groups.

**Source data 2.** Original files for gels of single-cell PCR are displayed in *Figure 4A*.

**Source data 3.** Data on the effect of 2-APB on the firing frequency of ventral tegmental area dopaminergic (VTA DA) neurons are displayed in *Figure 4B*.

**Source data 4.** Data on the effect of LA on the firing frequency of ventral tegmental area dopaminergic (VTA DA) neurons are displayed in *Figure 4C*.

**Source data 5.** Data on the effect of 2-APB on the resting membrane potential (RMP) of ventral tegmental area dopaminergic (VTA DA) neurons are displayed in *Figure 4D*.

**Source data 6.** Data on the effect of LA on the resting membrane potential (RMP) of ventral tegmental area dopaminergic (VTA DA) neurons are displayed in *Figure 4E*.

**Figure supplement 1.** Effects of TRPV channel blocker ruthenium red (RR) on the spontaneous firing of the male ventral tegmental area dopaminergic (VTA DA) neurons.

**Figure supplement 1—source data 1.** Data of the effects of ruthenium red (RR) on the spontaneous firing of the male ventral tegmental area dopaminergic (VTA DA) neurons.

the mRNA level of *Trpc6* in the VTA tissue (*Figure 5C*), and efficiency of viral infection into the VTA DA neurons (marked by DAT) was observed through visualization of GFP (*Figure 5D*). Both qPCR (*Figure 5C*) and immunofluorescence (*Figure 5D*) results indicated a sufficient repression of TRPC6 in the VTA DA neurons. Indicative of a significant role in the spontaneous firing and subthreshold depolarization, knockdown of TRPC6 (TRPC6-KD) resulted in a substantial reduction in the spontaneous firing frequency (0.43±0.07 Hz, n=37, N=12 for *Trpc6*- shRNA *vs* 1.88±0.16 Hz, n=27, N=11 for scramble-shRNA, Mann-Whitney U test, U=48, p<0.0001, for male and female mice, *Figure 5E*), and hyperpolarization of the RMP (−58.32±1.77 mV, n=10, N=6 for TRPC6-KD *vs* −50.24±0.65 mV, n=10, N=6 for Con, Mann-Whitney U test, U=8, p=0.0007, for male mice, *Figure 5G*) of the VTA DA neurons. In addition, the inhibitory effect of 2-APB and LA on the firing of VTA DA neurons infected with *Trpc6*-shRNA largely diminished, with no statistical difference in firing frequency before and after dosing (2-APB: from 0.58±0.15 Hz to 0.40±0.15 Hz, n=7, N=5, Wilcoxon matched-pairs signed rank test, W=−18.00, p=0.1563, for male mice, *Figure 5F*; LA: from 0.43±0.05 Hz to 0.43±0.06 Hz, n=8, N=5, Paired-sample t-test, t=0.000, df=7, 95% CI: −0.05740–0.05740, p>0.9999, for male mice, *Figure 5H*).

Consequence of knocking down TRPC6 described above should not be a result of a secondary effect on other TRPC channels with which TRPC6 is known to interact to form heteromers, such as TRPC4 and TRPC7 (*Strübing et al., 2003*; *Goel et al., 2002*), since in experiments using single-cell PCR (*Figure 5—figure supplement 1A*), it was found only a very small proportion of *Trpc6*-positive DA cells (*Trpc6+Slc6a3+*) expressed *Trpc4* (*Figure 5—figure supplement 1Bi*) or *Trpc7* (*Figure 5—figure supplement 1Bii*), in consistent with the results of single-cell RNA-seq results (*Figure 2*).

Taken together, the above results indicate that TRPC6 is an important contributor to subthreshold depolarization and spontaneous firing of the VTA DA neurons.

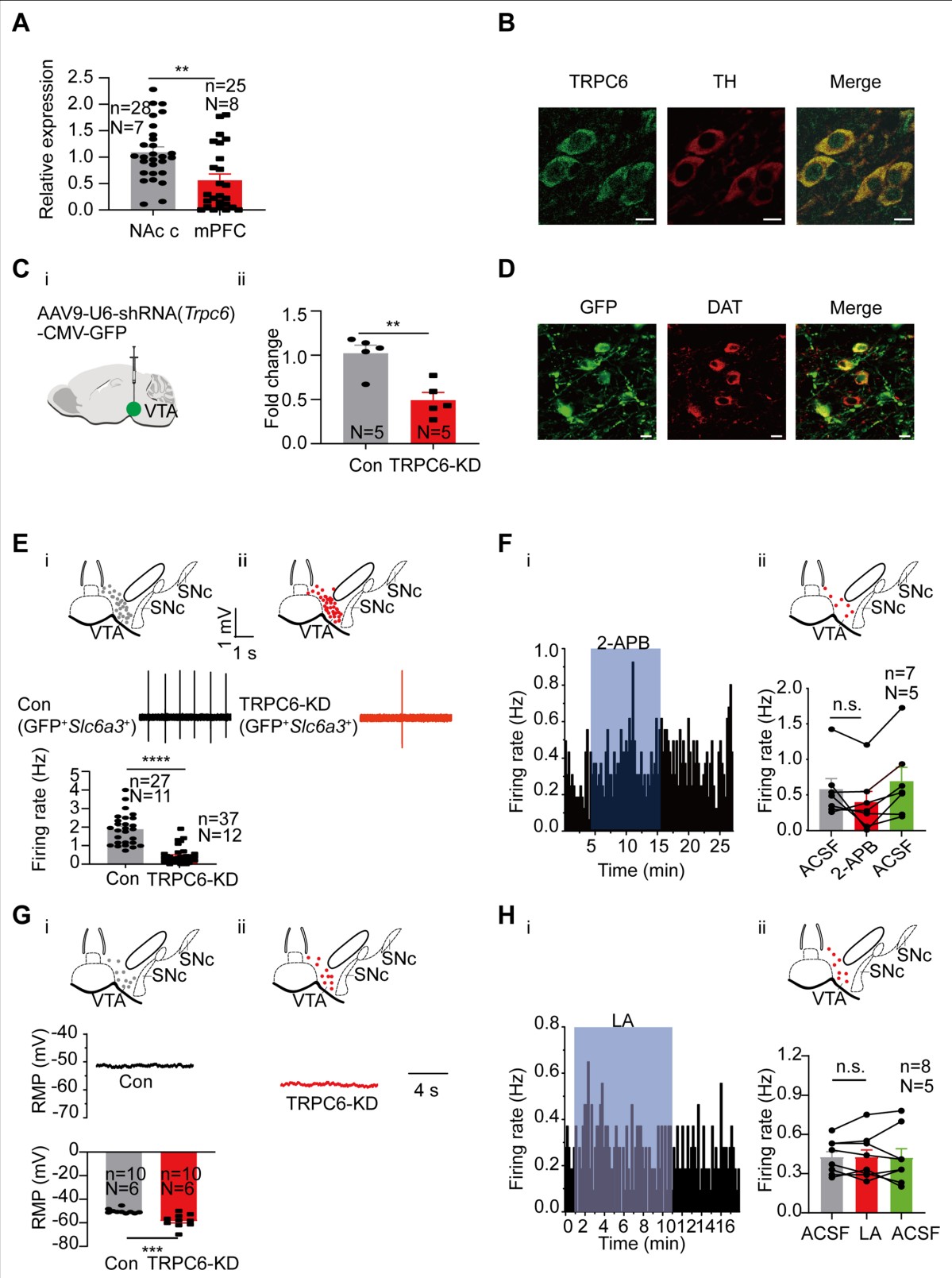

**Figure 5.** TRPC6 contributes to subthreshold depolarization and spontaneous firing of the ventral tegmental area dopaminergic (VTA DA) neurons. (**A**) The normalized expression profile of *Trpc6* in NAc c- and mPFC-projecting VTA DA neurons was verified using single cell-qPCR (NAc c: n=28, N=7; mPFC: n=25, N=8, both *Th* positive), Mann-Whitney U test, U=173, **p=0.0013. (**B**) Confocal images showing co-expression of TRPC6 (green) and TH (red) representing DA neurons (scale bar, 10 µm). (**C**) The efficiency of shRNA knockdown of *Trpc6* in the VTA was verified by qPCR. shRNA against *Trpc6*

*Figure 5 continued on next page*

*Figure 5 continued*

carried by AAV virus (**i**) (*Trpc6*-shRNA) was injected into the VTA of mice, the mRNA level in the shRNA-*Trpc6* transfected VTA (TRPC6-KD, N=5), and the scramble shRNA transfected VTA (Con, N=5) was analyzed using qPCR (**ii**). Two-sample t-test, t=4.223, df=8, 95% CI: –0.8194 to –0.2406, ** p=0.0029. (**D**) Immunofluorescence labeling showing the expression of AAV9-shRNA(*Trpc6*)-GFP (green) and DAT (red) in the VTA (scale bar, 10 μm). (**E**) Loose cell-attached current clamp recordings of the spontaneous firing of the VTA DA neurons from the mice of both sexes transfected with either *Trpc6*-shRNA (TRPC6-KD, **ii**, n=37, N=12) or scramble-shRNA (Con, **i**, n=27, N=11) (both GFP and *Slc6a3* positive). Examples of firing traces in the middle of (**i**) and (**ii**) and summarized data in the bottom of (**i**) were shown. The inset on the top of (**i**) and (**ii**) shows a map of a coronal midbrain slice indicating the location of GFP$^+$ neurons that were recorded and subsequently identified as DA neurons which were *Slc6a3* positive with single cell-PCR ((**i**) Con, gray dots; (**ii**) TRPC6-KD, red dots). Mann-Whitney U test, U=48, ****p<0.0001. (**F**) Example time-course (**i**) and summarized data (**ii**) showed that shRNA knockdown of *Trpc6* in the VTA of male mice decreased the 2-APB-inhibited firing responses of the VTA DA neurons. The inset on the top of (ii) shows a map of a coronal midbrain slice indicating the location of GFP$^+$ neurons that were recorded and subsequently identified as DA neurons which were *Slc6a3* positive with single cell-PCR (TRPC6-KD, red dots). (n=7, N=5, both GFP and *Slc6a3* positive), Wilcoxon matched-pairs signed rank test, W=–18.00, p=0.1563. (**G**). Whole-cell current clamp recordings of the resting membrane potential (RMP) of the male mice VTA DA neurons transfected with either *Trpc6*-shRNA (TRPC6-KD, (**ii**) n=10, N=6) or scramble-shRNA (Con, (**i**) n=10, N=6) (both GFP and *Slc6a3* positive). Examples of RMP traces in the middle of (**i**) and (**ii**) and summarized data in the bottom of (**i**) were shown. The inset on the top of (**i**) and (**ii**) shows a map of a coronal midbrain slice indicating the location of GFP$^+$ neurons that were recorded and subsequently identified as DA neurons which were *Slc6a3* positive with single cell-PCR ((**i**) Con, gray dots; (**ii**) TRPC6-KD, red dots). Mann-Whitney U test, U=8, ***p=0.0007. (**H**) Example time-course (**i**) and summarized data (**ii**) showed that shRNA knockdown of *Trpc6* in the male mice VTA decreased the LA-inhibited firing responses of the VTA DA neurons. The inset on the top of (**ii**) shows a map of a coronal midbrain slice indicating the location of GFP$^+$ neurons that were recorded and subsequently identified as DA neurons which were *Slc6a3* positive with single cell-PCR (TRPC6-KD, red dots). (n=8, N=5, both GFP and *Slc6a3* positive), Paired-sample t-test, t=0.000, df = 7, 95% CI: –0.05740–0.05740, p>0.9999. n.s. p>0.05, **p<0.01, ***p<0.001, ****p<0.0001. n is the number of neurons recorded and N is the number of mice used.

The online version of this article includes the following source data and figure supplement(s) for figure 5:

**Source data 1.** Data of the expression profile of *Trpc6* in NAc c- and medial prefrontal cortex (mPFC)-projecting ventral tegmental area dopaminergic (VTA DA) neurons by single cell-qPCR is displayed in the *Figure 5A*.

**Source data 2.** PDF file containing confocal images for *Figure 5B and D*.

**Source data 3.** Data of the expression level of *Trpc6* RNA in the shRNA-*Trpc6* transfected ventral tegmental area (VTA) and the scramble shRNA transfected VTA for *Figure 5C*.

**Source data 4.** Data of excitability of ventral tegmental area dopaminergic (VTA DA) neurons transfected with either *Trpc6*-shRNA or scramble-shRNA for *Figure 5E*.

**Source data 5.** Data of the effect of shRNA knockdown of *Trpc6* on the 2-APB-inhibited firing responses of the ventral tegmental area dopaminergic (VTA DA) neurons for *Figure 5F*.

**Source data 6.** Data of the effect of shRNA knockdown of *Trpc6* on the RMP of the ventral tegmental area dopaminergic (VTA DA) neurons for *Figure 5G*.

**Source data 7.** Data of the effect of shRNA knockdown of *Trpc6* on the LA-inhibited firing responses of the ventral tegmental area dopaminergic (VTA DA) neurons for *Figure 5H*.

**Figure supplement 1.** The expression of *Trpc4* and *Trpc7* channels in male ventral tegmental area (VTA) *Trpc6$^+$* dopaminergic (DA) neurons.

**Figure supplement 1—source data 1.** Data and a PDF file containing original gels for *Figure 5—figure supplement 1*, indicating the relevant bands and groups.

**Figure supplement 1—source data 2.** Original files for gels of single-cell PCR are displayed in *Figure 5—figure supplement 1*.

## Down-regulation of TRPC6 contributes to the altered firing activity of the VTA DA neurons in depression model

In multiple depression models, the depression-like behavior was directly linked to the altered firing activity of the VTA DA neurons (*Chaudhury et al., 2013*; *Friedman et al., 2014*; *Tye et al., 2013*; *Chang and Grace, 2014*; *Moreines et al., 2017*). In consideration of the evidence we described above that NALCN and TRPC6 play key roles in the firing activity of the VTA DA neurons, we went further to study if these channels also contributed to the altered firing activity of the VTA DA neurons and to the development of the depression-like behavior in depression models of mice.

We first established a mice depression model of chronic mild unpredictable stress (CMUS) (*Willner, 2017*), which, after 5 wk subjecting to two different stressors every day (*Figure 6A*), manifested chronic-stress-induced depression-like behaviors in the sucrose preference test (SPT) and the tail suspension test (TST), with reduced sucrose preference and lengthened immobility time, respectively (male mice: *Figure 6B*, female mice: *Figure 6—figure supplement 2B, I*). Multiple other behaviors tests were also performed on these CMUS male and female mice (male mice: *Figure 6—figure supplement 1*, female mice: *Figure 6—figure supplement 2A, C-H*), all indicating depression/anxiety-like behaviors.

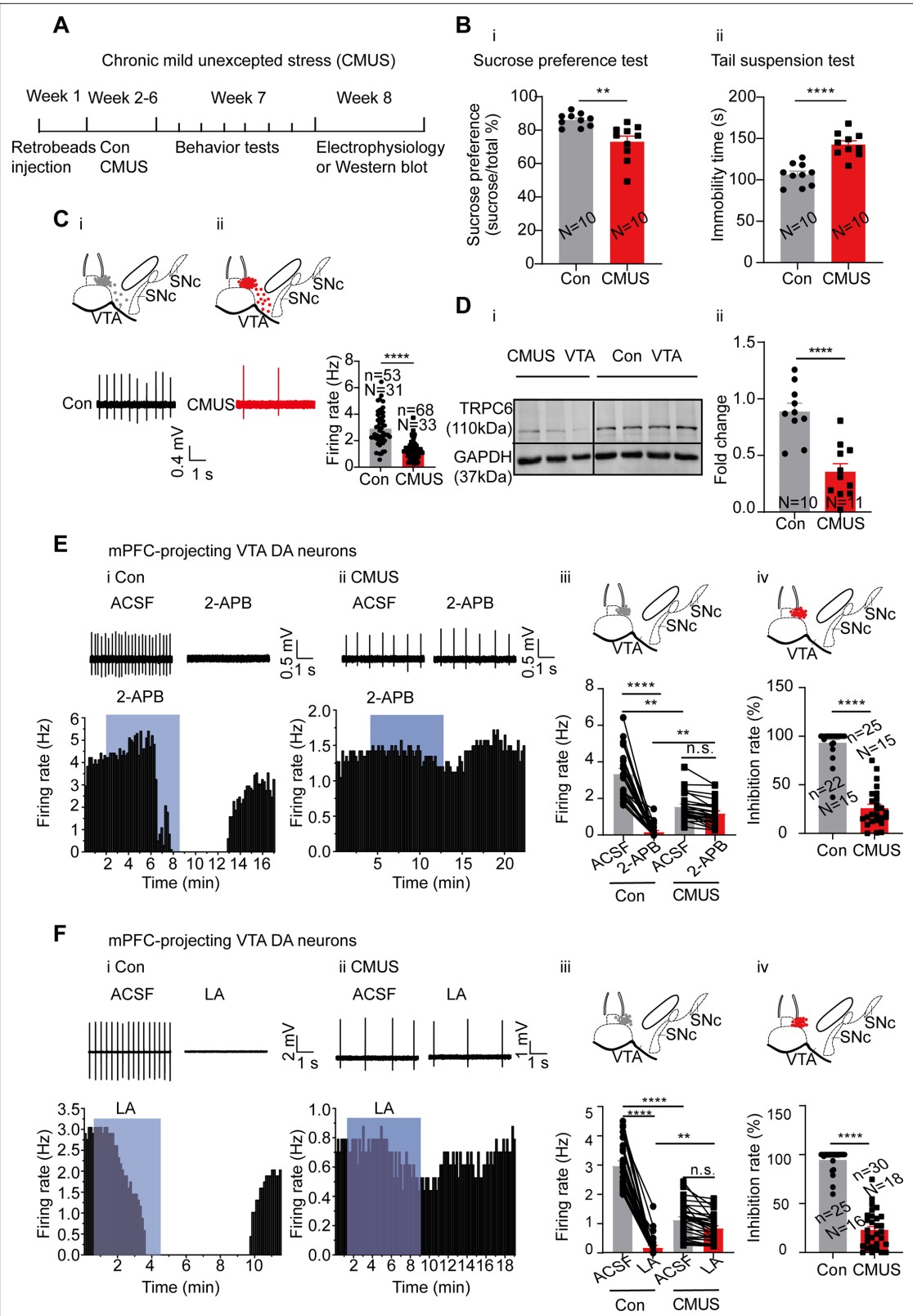

**Figure 6.** Chronic mild unpredictable stress (CMUS) depression mice have a decreased firing activity and down-regulated TRPC6 expression in the ventral tegmental area dopaminergic (VTA DA) neurons. (**A**) Experimental procedure timeline. (**B**) (**i**) Sucrose preference test for CMUS male mice (Con: N=10; CMUS: N=10). Two-sample t-test, t=3.569, df=18, 95% CI: –20.73 to –5.369, ** p=0.0022, compared with control mice. (**ii**) Tail suspension test for CMUS male mice (Con: N=10; CMUS: N=10), Two-sample t-test, t=5.826, df=18, 95% CI: 23.21–49.39, ****p<0.0001, compared with control mice.

*Figure 6 continued on next page*

*Figure 6 continued*

(**C**) Loose cell-attached current clamp recordings of the spontaneous firing of the VTA DA neurons from the CMUS (n=68, N=33, *Slc6a3* positive) and the control (n=53, N=31, *Slc6a3* positive) mice of both sexes. The inset on the top of (**i**) and (**ii**) shows a map of a coronal midbrain slice indicating the location of neurons that were recorded and subsequently identified as DA neurons which were *Slc6a3* positive with single cell-PCR ((**i**) Con, gray dots; (**ii**) CMUS, red dots). Examples of firing traces in the bottom of (**i**) as well as in the bottom left of (**ii**) and summarized data in the bottom right of (**ii**) were shown. Mann-Whitney U test, U=415, ****p<0.0001. (**D**) Representative Western blot assay (**i**) and summarized data (**ii**) showing the expression of TRPC6 and GAPDH in the VTA of the control (Con, N=10) and the CMUS (N=11) mice of both sexes. Two-sample t-test, t=5.134, df=19, 95% CI: –0.7480 to –0.3147, ****p<0.0001. (**E**) (**i**) and (**ii**) example traces (up) and time-course (bottom) of firing frequency of mPFC-projecting VTA DA neurons, the effect of 2-APB. (**i**) Con and (**ii**) CMUS of both sexes. (**iii**) Summarized effect of 2-APB in the control (n=22 cells, N=15, *Slc6a3* positive) and CMUS (n=25 cells, N=15, *Slc6a3* positive) mice. Kruskal-Wallis-H test with Dunnett's multiple comparisons test, Kruskal-Wallis statistic=66.03, Con-ACSF *vs.* CMUS-ACSF, **p=0.0056; Con-2-APB *vs.* CMUS-2-APB, **p=0.0016; Con-ACSF *vs.* Con-2-APB, ****p0.0001; CMUS-ACSF *vs.* CMUS-2-APB, p>0.9999. (**iv**) The inhibition rate (%) on firing rate by 2-APB in the control group (n=22, N=15) and the CMUS group (n=25, N=15). Mann-Whitney U test, U=7, ****p<0.0001. The inset on the top of (**iii**) and (**iv**) shows a map of a coronal midbrain slice indicating the location of mPFC-projecting neurons that were recorded and subsequently identified as DA neurons which were *Slc6a3* positive with single cell-PCR ((**iii**) Con, gray dots; (**iv**) CMUS, red dots). (**F**) (**i and ii**) Example traces (up) and time-course (bottom) of firing frequency of medial prefrontal cortex (mPFC)-projecting VTA DA neurons, effects of LA. (**i**) Con and (**ii**) CMUS of both sexes. (**iii**) Summarized effect of LA on the control (n=25 cells, N=16, *Slc6a3* positive) and CMUS (n=30 cells, N=18, *Slc6a3* positive) mice. Kruskal-Wallis-H test with Dunnett's multiple comparisons test, Kruskal-Wallis statistic = 79.59, Con-ACSF *vs.* CMUS-ACSF: ****p<0.0001, CON-LA *vs.* CMUS-LA: ** p=0.0026, Con-ACSF *vs.* Con-LA: ****p<0.0001, CMUS-ACSF *vs.* CMUS-LA: p>0.9999. (**iv**) The inhibition rate (%) on firing rate by LA in the control group (n=25, N=16) and the CMUS group (n=30, N=18). Mann-Whitney U test, U=2, ****p<0.0001. The inset on the top of (**iii**) and (**iv**) shows a map of a coronal midbrain slice indicating the location of medial prefrontal cortex (mPFC)-projecting neurons that were recorded and subsequently identified as DA neurons which were *Slc6a3* positive with single cell-PCR ((**iii**) Con, gray dots; (**iv**) CMUS, red dots). n.s. p>0.05, **p<0.01, ****p<0.0001. n is the number of neurons recorded and N is the number of mice used.

The online version of this article includes the following source data and figure supplement(s) for figure 6:

**Source data 1.** Data of the behavior test for chronic mild unpredictable stress (CMUS) male mice in *Figure 6B*.

**Source data 2.** Data of excitability of ventral tegmental area dopaminergic (VTA DA) neurons from the chronic mild unpredictable stress (CMUS) and the control mice for *Figure 6C*.

**Source data 3.** Data and a PDF file containing original western blots for *Figure 6D*, indicating the relevant bands and groups.

**Source data 4.** Original files for western blot analysis are displayed in *Figure 6D*.

**Source data 5.** Data of the effect of 2-APB on the firing frequency of medial prefrontal cortex (mPFC)-projecting ventral tegmental area dopaminergic (VTA DA) neurons between Con and chronic mild unpredictable stress (CMUS) for *Figure 6E*.

**Source data 6.** Data of the effect of LA on the firing frequency of medial prefrontal cortex (mPFC)-projecting ventral tegmental area dopaminergic (VTA DA) neurons between Con and chronic mild unpredictable stress (CMUS) for *Figure 6F*.

**Figure supplement 1.** Body weight and behavior of the male mice subjected to chronic mild unexpected stress (CMUS).

**Figure supplement 1—source data 1.** Data of body weight and behavior tests of the male mice between chronic mild unpredictable stress (CMUS) and Con.

**Figure supplement 2.** Body weight and behaviors of the female mice subjected to chronic mild unexpected stress (CMUS).

**Figure supplement 2—source data 1.** Data of body weight and behavior tests of the female mice between chronic mild unpredictable stress (CMUS) and Con.

**Figure supplement 3.** Expression of NALCN protein in the ventral tegmental area (VTA) of the chronic mild unpredictable stress (CMUS) male mice.

**Figure supplement 3—source data 1.** Data and a PDF file containing original western blots for *Figure 6—figure supplement 3*, indicating the relevant bands and groups.

**Figure supplement 3—source data 2.** Original files for western blot analysis are displayed in *Figure 6—figure supplement 3*.

**Figure supplement 4.** Body weight and behaviors of the male mice subjected to chronic restraint stress (CRS).

**Figure supplement 4—source data 1.** Data of body weight and behavior tests of the male mice between CRS and Con for *Figure 6—figure supplement 4B-I*.

**Figure supplement 4—source data 2.** Data and a PDF file containing original western blots for *Figure 6—figure supplement 4J*, indicating the relevant bands and groups.

**Figure supplement 4—source data 3.** Original files for western blot analysis are displayed in *Figure 6—figure supplement 4J*.

It has been shown that a reduced firing activity in the VTA DA neurons is responsible for the chronic-stress-induced depression-like behavior in the CMUS model (*Moreines et al., 2017*; *Liu et al., 2018*). Consistent with this finding, we also found the firing frequency of the VTA DA neurons from the CMUS male and female mice was significantly reduced as compared with that from the control male and female mice (2.90±0.17 Hz, n=53, N=31 for control mice vs 1.24±0.08 Hz, n=68, N=33 for CMUS

mice, Mann-Whitney U test, U=415, p<0.0001, *Figure 6C*). In agreement with a role for TRPC6 in this reduced firing activity, TRPC6 protein in the VTA tissue of male and female CMUS mice was found to be down-regulated in western blot experiments (*Figure 6D*). On the other hand, NALCN protein in the VTA was not altered in the CMUS male mice model of depression (*Figure 6—figure supplement 3*).

We also tested if this down-regulation of TRPC6 protein could also be found in a similar but different model of depression. For this, a chronic restraint stress (CRS) model was used. After 3 wk of restraint stress stimulation, the CRS male mice developed similar depression/anxiety-like behavior in the behavior tests like these similarly performed in the CMUS model (*Figure 6—figure supplement 4A–I*). Importantly, male mice with CRS-induced depression also showed a significant downregulation of TRPC6 protein in the VTA (0.38±0.06 fold, N=10 for control and N=14 for CRS, Two-sample t-test, t=7.369, df = 22, 95% CI: –0.8006 to –0.4489, p<0.0001, *Figure 6—figure supplement 4J*).

It is known that the activity of VTA dopamine neurons that project to mPFC is decreased in CMUS mice with depressive behaviors (*Zhong et al., 2018*; *Moreines et al., 2017*). Assuming that the decreased firing activity of the mPFC-projecting VTA DA neurons in the CMUS mice was associated with the down-regulation of TRPC6 expression, we reasoned that the firing activity in these CMUS mice would respond less to the TRP channel inhibitors. Indeed, TRP channel inhibitor 2-APB and TRPC6 inhibitor LA decreased their inhibitory effect on the firing activity of the mPFC-projecting VTA DA neurons from the CMUS male and female mice (2-APB: 1.56±0.15 Hz to 1.19±0.14 Hz, n=25, N=15, Kruskal-Wallis test, p>0.9999; LA: 1.12±0.11 Hz to 0.83±0.08 Hz, n=30, N=18, Kruskal-Wallis test, p>0.9999) compared with these from the control mice (2-APB: 3.35±0.29 Hz to 0.17±0.08 Hz, n=22, N=15, Kruskal-Wallis test, p<0.0001; LA: 2.97±0.16 Hz to 0.17±0.08 Hz, n=25, N=16, Kruskal-Wallis test, p<0.0001) (*Figure 6E and F*).

Taken together, the above results suggest down-regulation of TRPC6 plays a key role in the altered firing activity of the VTA DA neuron in mice models of depression.

## Down-regulation of TRPC6 in the VTA DA neurons confers the mice with depression-like behavior

If down-regulation of TRPC6 is crucial for the depression-like behaviors, like that indicated above in the CMUS and the CRS depression models, the down regulation of TRPC6 in non-stressfully treated mice should also develop depression-like behavior. We tested this possibility by selectively knocking down TRPC6 in the VTA DA neurons, using *Slc6a3-Cre* male mice injected with an AAV vector (AAV9-hSyn-DIO-shRNA(*Trpc6*)-RFP) (*Figure 7A*), driving the expression of shRNA against *Trpc6* selectively in the VTA DA neurons. Three weeks after the AAV injection, the qPCR (*Figure 7B*) and the immunofluorescence results (*Figure 7C*) indicated an efficient down-regulation of TRPC6 in the VTA DA neurons.

We next compared the depression- and anxiety-like behaviors in these conditionally TRPC6-knockdown male mice (TRPC6-cKD) with that in the control mice (infected with a virus carrying the scrambled shRNA). As shown in *Figure 7D*, in the TRPC6-cKD male mice, the sucrose preference was significantly reduced (7Di, 74.52±2.00%, N=10 for TRPC6-cKD, *vs* 85.87±1.85%, N=10 for control, Two-sample t-test, t=4.172, df=18, 95% CI: –17.07 to –5.634, p=0.0006), the immobility time in the Tail suspension test was significantly lengthened (7Dii, 142.00±4.66 s *vs* 121.40±3.94 s, Two-sample t-test, t=3.377, df=18, 95% CI: 7.785–33.41, p=0.0034). In the elevated plus maze test, the percentage of time spent in the open arm was significantly reduced (7Diii, 1.60±0.50% *vs* 11.42±2.04%, Mann-Whitney U test, U=2, p<0.0001) and the percentage of time spent in the closed arm was also significantly increased (7Div). And the Cre-dependent TRPC6-overexpression in male VTA *Slc6a3-Cre* mice (TRPC6-cOE) rescued the selective-TRPC6-knockdown-induced depression-like behaviors (*Figure 7E–H*). Furthermore, selective knockdown of TRPC6 in the mPFC-projecting VTA DA neurons (virus-retro carrying DIO-*Trpc6*-shRNA (TRPC6-rcKD) was injected into the mPFC of *Slc6a3-Cre* male mice) (*Figure 8A–C*) also conferred the mice with the depression- and anxiety-like behaviors (*Figure 8D*).

Taken together, these results suggest down-regulation of TRPC6 in the VTA DA neurons promotes the phenotypes of depression/anxiety of the mice in the CMUS.

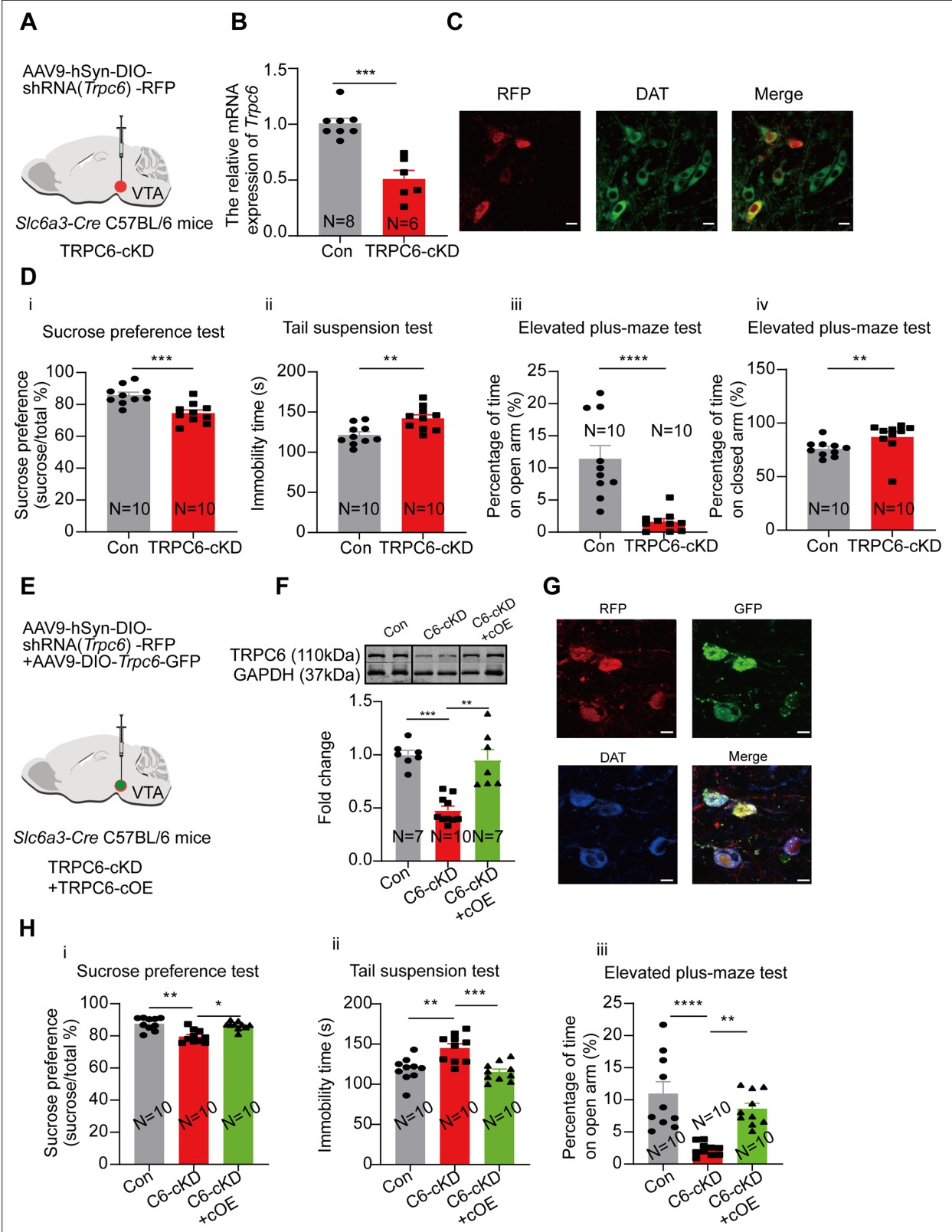

**Figure 7.** Selective knockdown of TRPC6 in the ventral tegmental area dopaminergic (VTA DA) neurons confers the male mice with depression-like and anxiety-like behaviors which are reversed by overexpression of TRPC6. (**A**) shRNA against *Trpc6* carried by AAV-DIO virus was injected into the VTA of the *Slc6a3-Cre* male mice. (**B**) The mRNA level in the shRNA-*Trpc6* transfected VTA (TRPC6-cKD) and the scramble shRNA transfected VTA (Con) was analyzed using qPCR (Con: N=8; TRPC6-cKD: N=6). Mann-Whitney U test, U=0, ***p=0.0003. (**C**) Immunofluorescence labeling showing the expression

*Figure 7 continued on next page*

*Figure 7 continued*

of AAV9-hSyn-DIO-shRNA(*Trpc6*)-RFP (red) and DAT (green) in the VTA of the *Slc6a3-Cre* mice (scale bar, 10 μm). (**D**) The effects of Cre-induced conditional knockdown of TRPC6 in the VTA DA neurons on the behaviors of mice in the sucrose preference test (**i**), the tail suspension test (**ii**), and the elevated plus-maze test (**iii, iv**) (Con: N=10; TRPC6-cKD: N=10). Sucrose preference test: Two-sample t-test, t=4.172, df=18, 95% CI: –17.07 to –5.634, ***p=0.0006; Tail suspension test: Two-sample t-test, t=3.377, df=18, 95% CI: 7.785–33.41, **p=0.0034; Elevated plus-maze test: Mann-Whitney U test, open arm, U=2, ****p<0.0001, closed arm, U=13, **p=0.0039. (**E**) AAV9-DIO-*Trpc6*-GFP (TRPC6-cOE) was injected into the VTA of the male *Slc6a3-Cre* mice, 7 d after injection of AAV9-hSyn-DIO-shRNA(*Trpc6*)-RFP (TRPC6-cKD). (**F**) The protein level TRPC6 of VTA (Con: the scramble shRNA for *Trpc6*, N=7; C6-cKD: TRPC6-cKD, N=10; C6-cKD+cOE: TRPC6-cKD+TRPC6 cOE, N=7). Kruskal-Wallis H test with Dunnett's multiple comparisons test, Kruskal-Wallis statistic=16.92, Con *vs.* C6-cKD: ***p=0.0009, C6-cKD *vs.* C6-cKD+cOE: **p=0.0034. (**G**) Immunofluorescence labeling showing the expression of AAV9-hSyn-DIO-shRNA(*Trpc6*)-RFP (red), AAV9-DIO-*Trpc6*-GFP (green) and DAT (blue) in the VTA of the *Slc6a3-Cre* mice (scale bar, 10 μm). (**H**) The effects of TRPC6 over-expression on depression-like behaviors of TRPC6-cKD male mice: the sucrose preference test (**i**), the tail suspension test (**ii**), and the elevated plus-maze test (**iii**) (Con: N=10; C6-cKD: N=10; C6-cKD+cOE: N=10). the sucrose preference test: Kruskal-Wallis H test with Dunnett's multiple comparisons test, Kruskal-Wallis statistic=12.96, Con *vs.* C6-cKD: **p=0.0025, C6-cKD *vs.* C6-cKD+cOE: *p=0.0139; tail suspension test: One-way ANOVA with Dunnett's multiple comparisons test, $F_{(DFn, DFd)}=0.8336_{(2, 27)}$, Con *vs.* C6-cKD: **p=0.0011, C6-cKD *vs.* C6-cKD+cOE: ***p=0.0003; elevated plus-maze test: One-way ANOVA with Dunnett's multiple comparisons test, $F_{(DFn, DFd)}=4.884_{(2, 27)}$, Con *vs.* C6-cKD: ****p<0.0001, C6-cKD *vs.* C6-cKD+cOE: **p=0.0023. *p<0.05, **p<0.01, ***p<0.001, ****p<0.0001. n is the number of neurons recorded and N is the number of mice used.

The online version of this article includes the following source data for figure 7:

**Source data 1.** Data of the relative mRNA level of *Trpc6* in the ventral tegmental area (VTA) between TRPC6-cKD and Con for the **Figure 7B**.

**Source data 2.** PDF file containing confocal images for **Figure 7C and G**.

**Source data 3.** Data of behavior tests between TRPC6-cKD and Con for **Figure 7D**.

**Source data 4.** Data and a PDF file containing original western blots for **Figure 7F**, indicating the relevant bands and groups.

**Source data 5.** Original files for western blot analysis are displayed in **Figure 7F**.

**Source data 6.** Data of the effects of TRPC6 over-expression on depression-like behaviors of TRPC6-cKD mice in **Figure 7H**.

## Discussion

In this study, we made a systematic study of the molecular mechanism for the subthreshold depolarization that drives the spontaneous firing of the VTA DA neurons. We identified TRPC6 channels, alongside NALCN, as major contributors to this subthreshold depolarization and related spontaneous firing, and further importantly, we also demonstrated that TRPC6 contributed to the altered firing activity of the VTA DA neurons under states of chronic-stress-induced depression-like behaviors.

The nature of a relatively depolarized resting membrane potential in the VTA DA neurons imposes a need for a full understanding of the channel conductance underlying this depolarization. It is first clear from these current and previous studies that this conductance is mediated by a persistent Na⁺ influx. Unlike the DA neurons in the SNc, where $Ca^{2+}$ influx is needed for the spontaneous firing (***Kang and Kitai, 1993***), the results in our and others' studies indicate that $Ca^{2+}$ influx is not necessary for the spontaneous firing of the VTA DA neurons. Replacement of $Ca^{2+}$ ions in the extracellular fluid with $Mg^{2+}$ accelerates the frequency of spontaneous firing (***Figure 1B***; ***Khaliq and Bean, 2010***), possibly resulting from a reduced activity of the $Ca^{2+}$-activated K⁺ currents and an increased activity of NALCN (***Philippart and Khaliq, 2018***; ***Lu et al., 2010***). Consistent with this, removing extracellular $Ca^{2+}$ did not hyperpolarize the RMP, which verifies more precisely for the role of $Ca^{2+}$ influx on the subthreshold depolarization. On the other hand, consistent with the previous study (***Khaliq and Bean, 2010***), a TTX-insensitive Na⁺ influx clearly contributed to the subthreshold depolarization (***Figure 1D***).

A TTX-insensitive nature of this Na⁺ conductance suggests that cation conductance permeable to Na⁺ might be the molecular candidate. We first focused on the HCN channels because, (1) it has been suggested HCN channels are important regulators for the spontaneous firing of VTA DA neurons in adult male mice under states of depression-like behaviors (***Friedman et al., 2014***; ***Zhong et al., 2018***); (2) *Hcn* are one of the more highly expressed NSCCs in the VTA DA neurons in our single-cell RNA-seq study; (3) HCN are permeable to Na⁺ (***Benarroch, 2013***). However, it is clear from our study that HCN does not contribute to the spontaneous firing of the VTA DA neurons in the adult male mice; none of the known HCN blockers we used had any significant effects on the firing activity. Presence of HCN has long been used as a signature of DA neurons (***Mercuri et al., 1995***), and also as a differentiating factor for the projection-specificity of the VTA DA neurons (***Lammel et al., 2008***). Nonetheless, the contribution of HCN to the spontaneous firing of the VTA DA neurons has been controversial; the HCN blockers ZD7288 and CsCl have been reported as both effective (***Seutin et al., 2001***; ***Neuhoff***

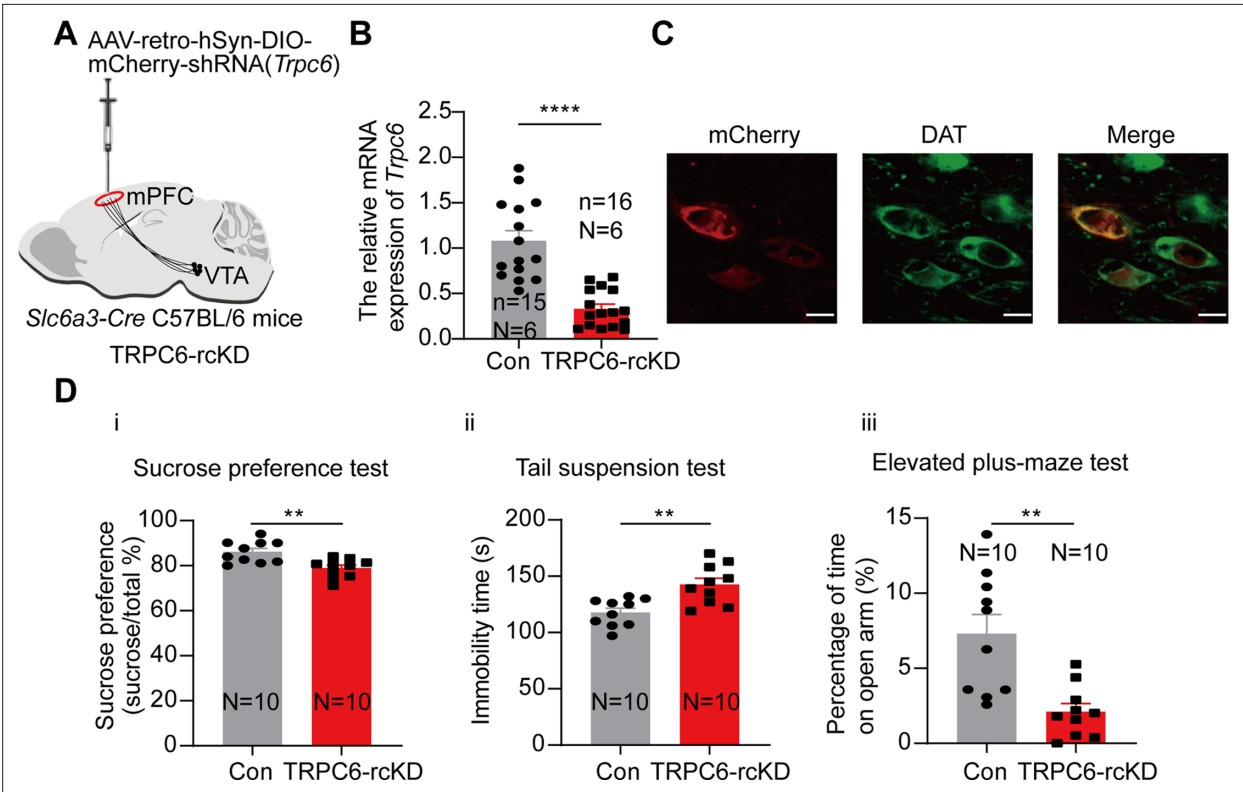

**Figure 8.** Selective knockdown of TRPC6 in the mPFC-projecting ventral tegmental area dopaminergic (VTA DA) neurons produces depression-like and anxiety-like behaviors. (**A**) shRNA against *Trpc6* carried by the retro-AAV-DIO vector was injected into the mPFC of the *Slc6a3-Cre* male mice. (**B**) The mRNA level in the shRNA-*Trpc6* transfected mPFC-projecting VTA (TRPC6-rcKD) DA neurons and the scramble shRNA transfected mPFC-projecting VTA DA neurons (Con) was analyzed using single-cell quantitative PCR (scqPCR) (Con: n=15, N=6; TRPC6-rcKD: n=16, N=6). Mann-Whitney U test, U=8.500, ****p<0.0001. (**C**) Immunofluorescence labeling showing the expression of AAV-retro-hSyn-DIO-mCherry-shRNA(*Trpc6*)(red) and dopamine transporter (DAT) (green) in the VTA of the *Slc6a3-Cre* mice (scale bar, 10 μm). (**D**) The effects of Cre-induced conditional knockdown of TRPC6 in mPFC-projecting VTA DA neurons on the behaviors of mice: (**i**) the sucrose preference test, (**ii**) the tail suspension test, and (**iii**) the elevated plus-maze test (Con: N=10; TRPC6-rcKD: N=10). Two-sample t-test, sucrose preference test: t=3.592, df=18, 95% CI: –11.48 to –3.008, **p=0.0021; tail suspension test: t=3.707, df=18, 95% CI: 10.79–39.01, **p=0.0016; elevated plus-maze test: open arm t=3.756, df=18, 95% CI: –8.099 to –2.288, **p=0.0014. **p<0.01, ****p<0.0001. n is the number of neurons recorded and N is the number of mice used.

The online version of this article includes the following source data for figure 8:

**Source data 1.** Data of the relative mRNA level of Trpc6 in the ventral tegmental area (VTA) mPFC-projecting dopaminergic (DA) neurons between TRPC6-rcKD and Con for the *Figure 8B*.

**Source data 2.** PDF file containing confocal images for *Figure 8C*.

**Source data 3.** Data of behavior tests for *Figure 8D*.

et al., 2002; Okamoto et al., 2006) and non-effective (Khaliq and Bean, 2010) on the spontaneous firing of the VTA DA neurons. A closer inspection of our and others' results presents a possible explanation for these different results: in the younger rodents (less than 1-mo-old Chan et al., 2007 and less than 15 d old in our study), the HCN in the DA neurons play a more important role whereas in the adult mice (~8 wk Chan et al., 2007), like the male adult ones (6~8 wk) we used in this study, the HCN does not contribute to the spontaneous firing of the VTA DA neurons. This is likely due to a shift of activation gating of HCN channels in a hyperpolarization direction with age, thus the modulation of HCN on the firing activity diminishes with age, as it is described in the SNc DA neurons (Chan et al., 2007). Consistent with this, we found hyperpolarizing the RMP rendered the VTA DA neurons of the adult male mice with sensitivity to the HCN blockers. Thus, the VTA DA neurons in adult male mice do not use HCN to depolarize the membrane to generate action potentials.

NALCN is widely expressed in central neurons (Lee et al., 1999; Lu and Feng, 2012), and is a popular candidate considered for the ion mechanism of the subthreshold depolarization currents and $Na^+$ leaky currents. NALCN has been shown to be an important component of background $Na^+$

currents and to be important for neuronal excitability in the hippocampal neurons, posterior rhomboid nucleus (Retrotrapezoid nucleus/RTN) chemo-sensitive neurons ($CO_2$/$H^+$-sensitive neurons), GABA neurons, and DA neurons in the substantia nigra, and spinal projection neurons (*Philippart and Khaliq, 2018*; *Shi et al., 2016*; *Lutas et al., 2016*; *Ford et al., 2018*; *Um et al., 2021*; *Lu et al., 2007*). Our results with single-cell RNA-seq, single-cell PCR, and immunofluorescence methods validated high-level expression of NALCN in the VTA DA neurons; both the pharmacological and the *Nalcn*-gene-knockout experiments demonstrated convincingly involvement of NALCN in subthreshold depolarization and spontaneous firing of the VTA DA neurons.

We did not detect alteration of NALCN protein expression in the depression model of CMUS in the VTA tissue. However, this does not completely exclude a possible contribution of NALCN to the altered functional activity of the VTA DA neurons which underlies the chronic-stress-induced depression-like behaviors. Future studies focusing on the functional property of NALCN in the cellular level of the VTA DA neurons under a state of depression are needed to clarify this issue. NALCN is after all a potential drug target in consideration of its important contribution to the resting membrane potential and functional activity of neurons.

The most interesting and important finding of this study is the role of TRPC6 in the regulation of the firing activity of the VTA DA neurons in both physiological and depression-state conditions; the experimental evidence from pharmacological, gene silencing, electrophysiological and behavior results unambiguously support this conclusion. It is interesting to note, although with an unclear neuronal circuit mechanism of action, the TRPC6 channel opener hyperforin was previously reported to have antidepressant effects in corticosterone-depressed mice, which were reversed by prior administration of the TRPC6 blocker larixyl acetate (*Pochwat et al., 2018*). Also, in a CMUS rat model, TRPC6 expression was found to be downregulated in the hippocampus, and administration of the TRPC6 opener hyperforin was effective in alleviating depression-like behavior (*Liu et al., 2015*). Our study provides strong and convincing evidence that TRPC6 is a key regulator of the VTA DA neurons which are known to be a key player in depression states (*Friedman et al., 2014*; *Li et al., 2017*; *Tye et al., 2013*; *Chang and Grace, 2014*; *Cao et al., 2010*).

The above-mentioned role for TRPC6 should come from its contribution to the subthreshold depolarization, through a persistent permeation to the influx of $Na^+$. Although TRPC channels have been better recognized for their permeability to $Ca^{2+}$ and their regulation of cellular $Ca^{2+}$ homeostasis, the importance of TRPC channels for the permeability of monovalent cations, especially $Na^+$, is now gradually being appreciated (*Eder et al., 2005*). In the substantia nigra GABA and DA neurons, TRPC3 has been reported to play an important role in maintaining depolarization membrane potential, peace-making, and firing regularity (*Um et al., 2021*; *Zhou et al., 2008*). Otherwise, it has been reported that the slow tonic firing of SNc DA neurons, depends on the basal activity of both the NALCN and TRPC3 channels, but that burst firing does not require TRPC3 channels but relies only on NALCN channels (*Hahn et al., 2023*). TRPC6 and TRPC3 belong to the same TRPC family subclass and TRPC6 is more than 75% homologous to TRPC3 (*Clapham et al., 2001*). Interestingly, our results from single-cell RNA-seq and single-cell PCR demonstrated the rich expression of *Trpc6* in the VTA DA neurons among all TRP channels but no expression of *Trpc3* was found in these neurons. These results present an opportunity for selectively targeting a single TRPC channel to exert selective pharmacological results. Besides, TRPC4 expression was detected in a uniformly distributed subset of rat VTA DA neurons (*Klipec et al., 2016*), which differs with our results with single-cell RNA-seq and single-cell PCR, and TRPC4 KO rats showed decreased VTA DA neuron tonic firing and deficits in cocaine reward and social behaviors (*Klipec et al., 2016*).*Trpc4* is detected to be expressed with a low level in the VTA DA neurons of male mice by single-cell RNA-seq (*Figure 2*) and in *Trpc6* and *Slc6a3* double-positive VTA cells by single-cell PCR (*Figure 5—figure supplement 1*). The different species used in this and our study may be one explanation, but the role of TRPC4 in the excitability of mice VTA TRPC6-negative DA neurons is worth to investigating further.

It has also been demonstrated in cell types other than neurons, such as rat portal vein smooth muscle cells, that the activation of the α-adrenergic receptor can open TRPC6, which leads to cell depolarization, activation of voltage-dependent $Ca^{2+}$ channels, and ultimately the contraction of smooth muscle cell (*Inoue and Mori, 2002*); it is also shown that angiotensin II and endothelin I can activate TRPC3/C6 heterodimers via vascular G protein-coupled receptors (GPCR), thereby depolarizing cells (*Nishida et al., 2010*; *Nishioka et al., 2011*). Following this line, it would be very interesting

to know if TRPC6 in the VTA DA neurons, where multiple GPCR reside receiving input and auto modulation by neuronal transmitters (*Su et al., 2019*; *McCall et al., 2019*), could also be a target of modulation with a similar mechanism. In consideration of the important role, we described here in this study, this type of modulation will present possible explanations for some unsolved mechanisms for physiological and pathophysiological functions of the VTA DA neurons.

The facts that down-regulation of TRPC6 proteins was correlated with reduced firing activity of the VTA DA neurons as well as the chronic-stress-induced depression-like behaviors of mice with both sexes, and that knocking down of TRPC6 in the VTA DA neurons confers the male mice with depression-like behaviors strongly suggest a crucial role for TRPC6 in the development of chronic-stress-induced depression-like behaviors under stressful conditions like CMUS. To reinforce this, down-regulation of TRPC6 was also found in another chronic-stress-induced depression model of CRS, a model with similar neuronal alteration of reduced firing activity in the VTA DA neurons to that described in the CMUS model (*Qu et al., 2020*). These facts with the fact that TRPC6 activator hyperforin is an effective antidepressant in multiple depression models (*Pochwat et al., 2018*; *Liu et al., 2015*), and that hyperforin is the principal component of St. John's wort, a well-known antidepressant herb (*Butterweck, 2003*), present TRPC6 as a very attractive drug target for new lines of antidepressants.

## Materials and methods

### Animal preparation

Male or female 6–8 wk-old C57BL/6 (Vital River, China) and *Slc6a3-Cre* mice with a C57BL/6 background (Stock No: 006660, the Jackson Laboratories, USA) were used for the studies. Ethics Statement: All experiments were conducted in accordance with the guidelines of the Animal Care and Use Committee of Hebei Medical University and approved by the Animal Ethics Committee of Hebei Medical University (Shijiazhuang, China), 2017013.

### Brain slice preparation

The details for the preparation of coronal brain slice containing VTA were the same as described in our previously published work (*Li et al., 2017*; *Su et al., 2019*). Briefly, mice were anesthetized with chloral hydrate (200 mg/kg, i.p.). After intracardial perfusion with ice-cold sucrose solution, the brains of the mice were removed quickly and placed into the slicing solution. The ice-cold sucrose-cutting solution contained (in mM): sucrose 260, NaHCO$_3$ 25, KCl 2.5, NaH$_2$PO$_4$ 1.25, CaCl$_2$ 2, MgCl$_2$ 2, and D-glucose 10; osmolarity, 295–305 mOsm. Using the vibratome (VT1200S; Leica, Germany), the coronal midbrain slices (250 µm-thick) containing VTA were sectioned. The slices were incubated for 30 min at 36 °C in oxygenated artificial cerebrospinal fluid (ACSF) (in mM: NaCl 130, MgCl$_2$ 2, KCl 3, NaH$_2$PO$_4$ 1.25, CaCl$_2$ 2, D-Glucose 10, NaHCO$_3$ 26; osmolarity, 280–300 mOsm), and were then left for recovery for 60 min at room temperature (23–25°C) until use.

The brain slices were transferred to the recording chamber and were continuously perfused with fully oxygenated ASCF during recording.

### Identification of DA neurons and electrophysiological recordings

Recordings in the slices were performed in whole-cell current-clamp and voltage-clamp configurations on the Axoclamp 700B preamplifier (Molecular Devices, USA) coupled with a Digidata 1550B AD converter (Molecular Devices, USA). Neurons in the VTA were visualized with a 40x water-immersion objective equipped by an optiMOS microscope camera (Qimaging, Canada) on an Olympus-BX51 microscope (Olympus, Japan). Projection-specific or GFP-positive VTA neurons were identified by infrared-differential interference contrast (IR-DIC) video microscopy and epifluorescence (Olympus, Japan) for detection of retrobeads (red) positive or GFP-positive neurons.

For whole-cell recording, glass electrodes (3–5 MΩ) were filled with internal solution (in mM): K-methylsulfate 115, KCl 20, MgCl$_2$ 1, HEPES 10, EGTA 0.1, MgATP 2 and Na$_2$GTP 0.3, pH adjusted to 7.4 with KOH. And the extracellular solution was the ACSF.

In whole-cell current-clamp mode, the HCN function was judged by the inwardly rectifying characteristic sag potential generated by giving a hyperpolarizing current (–100 pA).

When recording the effect of removing extracellular $Ca^{2+}$ on spontaneous cell discharge, $CaCl_2$ in the ACSF was replaced with $MgCl_2$, and the final $MgCl_2$ concentration was 4 mM.

The resting membrane potential (RMP) was measured in current clamp mode (I=0); the composition of the recording solution (mM) contained: NaCl 151, KCl 3.5, $CaCl_2$ 2, $MgCl_2$ 1, glucose 10, and HEPES 10, and the pH was adjusted to about 7.35 with NaOH. 1 µM tetrodotoxin was added to the extracellular solution to abolish action potential during measurement of the RMP.

To observe the effect of extracellular $Na^+$ on the RMP, NaCl in the original recording solution was replaced with equimolar NMDG, and the NMDG-recording solution (mM) contained: NMDG 151, KCl 3.5, $CaCl_2$ 2, $MgCl_2$ 1, glucose 10, and HEPES 10, and the pH was adjusted to 7.35 with KOH.

For recording the spontaneous firing of the neurons, cell-attached 'loose-patch' (100–300 MΩ) recordings were used (*Burlet et al., 2002*). In this case, patch-pipettes (2–4 MΩ) were filled with ACSF, and the spontaneous activity was recorded in the current-clamp mode (I=0). The synaptic blockers (CNQX, 10 µM; APV, 50 µM and gabazine, 10 µM) were added to isolate the intrinsic firing properties.

At the end of electrophysiological recordings, the recorded VTA neurons were collected for single-cell PCR. VTA DA neurons were identified by single-cell PCR for the presence of *Th* and *Slc6a3*.

## Retrograde labeling

Briefly, under general chloral hydrate anaesthesia (200 mg/kg, i.p.) and stereotactic control (RWD Instruments, Guangzhou, China), the skull surface was exposed. All coordinates are relative to bregma in mm using landmarks and neuroanatomical nomenclature that was described in the Franklin and Paxinos mouse brain atlas (*Paxinos and Franklin, 2001*). Red retrobeads (100 nl for single injection; Lumafluor Inc, Naples, FL, USA) were injected into the following sites by KD scientific syringe pump (KD scientific, Holliston MA, USA): bilaterally into NAc core (NAc c), NAc lateral shell (NAc ls), NAc medial shell (NAc ms) and basolateral amygdala (BLA), four separate sites (2 per hemisphere) into medial prefrontal cortex (mPFC). Coordinates for infusions were as follows: NAc c (AP +1.50, ML ±0.84, DV –4.00~–3.80; 100 nl beads); NAc ls (AP +0.86, ML ±1.72, DV –4.24~–3.93; 100 nl beads); NAc ms (AP +1.70, ML ±0.53, DV –4.16~–3.60; 100 nl beads); BLA (AP –1.46, ML ±2.85, DV –4.01~–3.58; 100 nl beads); mPFC (AP +2.05 and 2.15, ML ±0.27, DV –2.26~–1.70; 200 nl beads). Retrobeads were delivered through a pulled glass pipette using a PAP107 Multi-pipette Puller (MicroData Instrument, Inc, USA) and at a rate of 100 nl/min; the injection needle was left in place for at least 5 min after each infusion. Following surgery, mice were returned to a single housing. For sufficient labeling, survival periods for retrograde tracer transport depended on respective injection areas: NAc c, NAc ls, NAc ms, 14 d; mPFC, 21 d; BLA 14 d (*Lammel et al., 2008*). Coronal sections of injection sites were stained with 4, 6-diamidino-2-phenylindole (DAPI, Sigma, USA) to confirm representative target location. Then, serial analyses of the injection-sites were carried out routinely.

## AAV for gene knockdown or overexpression and viral construct and injection

For knockdown of NALCN and TRPC6, AAV9-U6-shRNA(*Nalcn*)-CMV-GFP (300 nl) or AAV9-U6-shRNA(*Trpc6*)-CMV-GFP or its control AAV9-scramble-shRNA was delivered into the VTA (AP –3.08 mm; ML ±0.50 mm; DV –5.00~–4.20 mm) of the mice. AAV9-hSyn-DIO-shRNA(*Trpc6*)-RFP (300 nl) or its control AAV9-scramble-shRNA was delivered into the VTA of the *Slc6a3-Cre* C57BL/6 mice. AAV-retro-hSyn-DIO-mCherry-shRNA(*Trpc6*) (300 nl) or its control AAV-retro-scramble-shRNA was delivered into the mPFC of the *Slc6a3-Cre* C57BL/6 mice.

The shRNA hairpin sequences used in this study:

> *Nalcn* shRNA: 5'-AAGATCGCACAGCCTCTTCAT-3'(*Shi et al., 2016*);
> *Trpc6* shRNA: 5'-CCAGGATCAATGCATACAA-3'.

For Cre-dependent overexpression of TRPC6, AAV9-DIO-*Trpc6*-GFP (300 nl) or its control AAV9-DIO-GFP was delivered into the VTA of the *Slc6a3-Cre* C57BL/6 mice, 7 d after injection of AAV9-hSyn-DIO-shRNA(*Trpc6*)-RFP.

Mice were singly housed with enough food and water to recover for 4–5 wk after injection of the virus, before behavior tests and electrophysiological recordings.

## Single-cell PCR

mRNA was reversely transcribed to cDNA by PrimeScript II 1st Strand cDNA Synthesis Kit (Takara-Clontech, Kyoto, Japan). At the end of electrophysiological recordings, the recorded neuron was aspirated into a pipette and then expelled into a PCR sterile tube containing 1 μl oligo-dT Primer and 1 μl dNTP mixture. The mixture was heated to 65 °C for 5 min to denature the nucleic acids and then cooled on ice for 2 min. Reverse transcription from mRNA into cDNA was performed at 50 °C for 50 min and then 85 °C for 5 min. cDNA was stored at –40 °C. Then two rounds of conventional PCR with pairs of gene-specific outside (first round) and inner primers (second round) for *Gapdh* (positive control), *Th*, *Slc6a3*, *Drd2*, *Kcnj6*, *Slc17a6*, *Gad1*, *Nalcn*, *Trpc6*, *Trpv2*, *Trpc3*, *Trpc4* and *Trpc7* using GoTaq Green Master Mix (Promega, Madison, USA) were performed. After adding the specific outside primer pairs into one PCR tube per cell, first-round synthesis conditions were as follows: 95 °C (5 min); 30 cycles of 95 °C (50 s), 58–63°C (50 s), 72 °C (50 s); 72 °C (5 min). Then, the product of the first PCR was added in the second amplification round by using a specific inner primer (final volume 25 μl). The second amplification round consisted of the following: 95 °C (5 min); 35 cycles of 95 °C (50 s), 58 °C-62 °C (50 s), 72 °C (50 s) and 5 min elongation at 72 °C. The final PCR products were separated by electrophoresis on 2% agarose gels. The negative control reactions with no added cDNA were also performed in each experiment.

The 'outer' primers (from 5′ to 3′) are as follows:

Gapdh:
AAATGGTGAAGGTCGGTGTGAACG (sense)
AGTGATGGCATGGACTGTGGTCAT (antisense)
Th:
GCCGTCTCAGAGCAGGATAC
GGGTAGCATAGAGGCCCTTC
Slc6a3:
CTGCCCTGTCCTGAAAGGTGT
GCCCAGTGATCACAGACTCC
Drd2:
AGCATCGACAGGTACACAGC
CCATTCTCCGCCTGTTCACT
Kcnj6:
TGGACCAGGATGTGGAAAGC
AAACCCGTTGAGGTTGGTGA
Slc17a6:
CTGCTTCTGGTTGTTGGCTACTCT
ATCTCGGTCCTTATAGGTGTACGC
Gad1:
ACAACCTTTGGCTGCATGTGGATG
AATCCCACGGTGCCCTTTGCTTTC
Nalcn:
TCCATCTGTGGGAAGCATGT
CAAAAGCTGGTCCTCTTCAGTG
Trpc6:
AGCCTGTCTATTGAGGAAGAAC
AGCGAGAATGATTGGGGTCA
Trpv2:
CTGCACATCGCCATAGAGAA
AGGCTGGTGGTAGGCAACTA
Trpc3:
GCTGGCCAACATAGAGAAGG
CCTGCACGTGACTATCCACA
Trpc4:
GTCTATGTAGGCGATGCGCT
AGGAGGTCCTTGGCAAATTGT
Trpc7:

GCCATCAGCAAGGGCTATGT
CACGCCCACCACAAAATCC

The 'inner' primers (from 5' to 3') are as follows:

Gapdh:
GCAAATTCAACGGCACAGTCAAGG
TCTCGTGGTTCACACCCATCACAA
Th:
AGGAGAGGGATGGAAATGCT
ACCAGGGAACCTTGTCCTCT
Slc6a3:
ATTTTGAGCGTGGTGTGCTG
TGCCTCACAGAGACGGTAGA
Drd2:
CCATTGTCTGGGTCCTGTCC
GTGGGTACAGTTGCCCTTGA
Kcnj6:
AGCCGAGACAGGACCAAAAG
ATGTACGCAATCAGCCACCA
Slc17a6:
CATCTCCTTCTTGGTGCTTGCAGT
ACAGCGTGCCAACGCCATTTGAAA
Gad1:
AGTCACCTGGAACCCTC
GCTTGTCTGGCTGGAA
Nalcn:
GCCTTTGCTGGAGTTGTTCTG
CCCTGCATAATTGCCACAGTC
Trpc6:
TGCTAGAAGAGTGTCATTCCCT
CCTCCACAATCCGTACATAACC
Trpv2:
GTGGGATGTGGTGACCTACC
GCTGGTACAGCCCTGAGAAC
Trpc3:
CCTTGGGTCTTCCATTCCTC
CACAACTGCACGATGTACTCC
Trpc4:
CCACGAGGTCCGCTGTAAC
CTCCCAACTTAACTGAAAGGCA
Trpc7:
CGCTTCTCCCACGACATCAC
ACTGGATAGGGACAGGTAGGC
RT-qPCR:
Total RNA was prepared using Trizol. RNA was reversely transcribed into cDNA by TAKARA PrimeScript RT reagent Kit with gDNA Eraser for Reverse transcription. Subsequently, the TB Green Premix Ex Taq II (Tli RNaseH Plus) (TaKaRa) was used to do the RT-qPCR assays.

The specific primers were:

*Gapdh* :
GCAAATTCAACGGCACAGTCAAGG
TCTCGTGGTTCACACCCATCACAA
*Trpc6* :
GCAGAAACACAGAGGAAGTGG
GCTCTTTCCAGCTTGGCATATC

*Nalcn* :
TAATGAGATAGGCACGAGTA
TGATGAAGTAGAAGTAGGAG

## Single-cell RT-qPCR

Following the retrograde labeling and VTA brain slice preparation, projection-specific VTA neurons were identified by infrared-differential interference contrast (IR-DIC) video microscopy and epifluorescence (Olympus, Japan) for detection of retrobeads (red) positive neurons. mRNA was reversely transcribed to cDNA by PrimeScript II1st Strand cDNA Synthesis Kit (Takara-Clontech, Kyoto, Japan). The projection-specific neuron was aspirated into a pipette and then expelled into a PCR sterile tube containing 1 µl oligo-dT Primer and 1 µl dNTP mixture. The mixture was heated to 65 °C for 5 min to denature the nucleic acids and then cooled on ice for 2 min. Reverse transcription from mRNA into cDNA was performed at 50 °C for 50 min and then 85 °C for 5 min. cDNA was stored at –40 °C. The targeted pre-amplification was done, with pairs of targeted primers for *Gapdh*, *Th*, and *Trpc6*, to quantify multiple targets per cell. After adding the targeted primer pairs into each PCR tube, the synthesis conditions were as follows: 95 °C (5 min); 10 cycles of 95 °C (50 s), 58–62°C (50 s), 72 °C (50 s); 72 °C (5 min). After targeted pre-amplification, quantitative PCR (qPCR) is the final laboratory step of the scRT-qPCR workflow with the TB Green Premix Ex Taq II (Tli RNaseH Plus) (TaKaRa).

The targeted primers for pre-amplification (from 5'to 3') are as follows:

Gapdh:
AAATGGTGAAGGTCGGTGTGAACG (sense)
AGTGATGGCATGGACTGTGGTCAT (antisense)
Th:
AAGGTTCATTGGACGGCGG
ACATCGTCAGACACCCGAC
Trpc6:
AGCCTGTCTATTGAGGAAGAAC
AGCGAGAATGATTGGGGTCA
The primers for qPCR (from 5' to 3') as follows:
Gapdh:
GCAAATTCAACGGCACAGTCAAGG
TCTCGTGGTTCACACCCATCACAA
Th:
TCTCCTTGAGGGGTACAAAACC
ACCTCGAAGCGCACAAAGT
Trpc6:
TGCTAGAAGAGTGTCATTCCCT
CCTCCACAATCCGTACATAACC

## Immunofluorescence

After intracardial perfusion with 4% paraformaldehyde (PFA) in 0.01 M PBS (pH 7.4), the brains were post-fixed in 4% paraformaldehyde. 48 hr later, the brain tissue was placed in 30% sucrose solution (PBS preparation) to dehydrate. The brain tissue was sectioned coronally, including the NAc, mPFC, BLA, and VTA, using a vibrating microtome. The section thickness of NAc, mPFC, and BLA was 80 µm and the section thickness of VTA was 40 µm. The sections were placed in PBS solution and stored in a refrigerator at 4 °C for storage.

The brain section was incubated in 0.3% triton/3% BSA for 1 hr at room temperature and then was blocked with 10% donkey serum at 37 °C for 1 hr. After that, the brain section was incubated in the corresponding antibodies in PBS at 4°C for 12 hr. The section was washed three times (10 min) with PBS. Finally, the brain section was incubated in the secondary antibodies for 2 hr at 37 °C. Images were obtained on a Leica TCS SP5 confocal laser microscope (Leica, Germany) equipped with laser lines for 405 mm, 488 mm, 561 mm, and 647 mm illumination. Images were analyzed with LAS-AF-Lite software (Leica, Germany).

### Single-cell whole-transcriptome gene sequencing

After retrobeads injection, following brain slice preparation and recording, the recorded and retrobeads-labeled neurons were aspirated into the patch pipette and were then broken into the PCR tube containing 1 µl lysis buffer. For mRNA in individual cells, mRNA was amplified by SMARTer Ultra Low Input RNA for Illumina Kit, which was qualified and reversely transcribed to cDNA by Qubit and Agilent Bioanalyzer 2100 electrophoresis. After the fragmentation of cDNA (300 bp) by ultrasound, sequencing libraries (end repair, addition of poly(A), and ligation of sequencing connectors) were built using the Ovation Ultralow Library System V2. After that, the constructed libraries were sequenced using Illumina HiseqXten. The dataset is stored in NCBI Gene Expression Omnibus https://www.ncbi.nlm.nih.gov/geo/query/acc.cgi?acc=GSE276319, and in DRYAD https://doi.org/10.5061/dryad.41ns1rnq5.

### Depression models

#### CMUS procedure

The CMUS procedures consisted of food and water deprivation (24 hr), day/night inversion, damp bedding (12 hr), cage tilt (12 hr), no bedding (12 hr), rat bedding (12 hr), 4 °C cold bath (5 min), restraint (2 hr) and tail pinching (30 min). Mice were subjected to consecutive 35 d of CMUS with two stressors per day. Non-stressed controls were handled only for cage changes and behavioral tests.

#### CRS procedure

The mice were immobilized in a special restraint device for 6 hr per day from 9:00 to 15:00 for 21 d.

#### Behavior tests

The behavior tests were carried out by experimenters blind to the group, in the following order.

#### Sucrose preference test

Mice were single-housed and trained to drink from two drinking bottles with 1 bottle of tap water and 1 bottle of 1.0% sucrose water, for 48 hr, and the positions of the drinking bottles were exchanged every 12 hr to exclude the interference of position preference. After the training, the mice were deprived of food and water for 12 hr, and then the sucrose preference test was performed for 24 hr. One bottle of tap water and one bottle of 1.0% sucrose water were given again, and the positions of the drinking bottles were exchanged at 12 hr. Finally, the consumption of tap water and sugar water at 24 hr was recorded to calculate the sucrose solution preference rate. Sucrose preference rate (%)=sucrose solution consumption/(tap water consumption + sucrose solution consumption)*100%.

#### Open field test, OFT

The mice were placed in a topless chamber (40 cm×40 cm×30 cm) with a camera mounted on top of the chamber and connected to an ANY-maze video tracking system on the computer to automatically track and record the activity of mice during the experiment and to obtain behavioral data. The bottom of the chamber was divided into 16 small square grids (10 cm×10 cm) on the computer, and four square grids in the central area were defined as the central zone. During the 10 min test session, the total distance traveled and time spent in the center were recorded.

#### Elevated plus maze, EPM

The EPM apparatus consists of two closed arms (25×5 cm) across from each other and perpendicular to two open arms (25×5 cm) that are connected by a center platform (5×5 cm), with a camera in the center ceiling to automatically track and record the activity of mice. Mice were placed in the center platform facing a closed arm and allowed to freely explore the maze for 7 min, of which the first 2 min was the adaptation time. The time spent in open and closed arms was analyzed for the last 5 min after the adaptation.

## Accelerating rotarod test

Locomotor activity was estimated by accelerating the rotarod test. Each group of animals was placed on an accelerating rotarod walking for 300 s from 4 to 40 rpm. After 3 d of training (three times per day), latency was recorded from the beginning of the trial until the mouse falls off for 300 s from 4 to 40 rpm.

## Forced swimming test, FST

FST was performed in a glass cylinder, filled with the water depth of about 15 cm at room temperature. The experiment was conducted for 5 min, of which the first 1 min was the adaptation time and the mice were allowed to move freely, and the immobility time was recorded for the last 4 min after the adaptation.

## Tail suspension test, TST

Mouse was suspended by taping the tail (1 cm from the tip of the tail) to the fixation device for 6 min. The cumulative immobility time within the last 4 min was recorded, and the time when all limbs were immobile except for respiration was considered as immobility time.

## Western blot

VTA was carefully obtained from the coronal midbrain slices (900 µm-thick) with a small homemade biopsy punch (1.0 mm in diameter) on ice, which was verified using the Franklin and Paxinos mouse brain atlas (*Paxinos and Franklin, 2001*), and then stored at –80 °C until the day of the experiment. The VTA protein was isolated by 100 µl RIPA and 1 µl PMSF. The total protein for each sample was transferred onto the polyvinylidene difluoride (PVDF) membranes after SDS-polyacrylamide gel electrophoresis (SDS-PAGE), and blocked 2 hr at room temperature with 5% bovine serum albumin (KeyGen Biotechnology). Subsequently, all these membranes were incubated overnight at 4 °C with the primary antibodies as below: TRPC6, NALCN, and GAPDH. Following 2 hr secondary antibodies incubation, all the bands were detected. The relative expression was calculated based on the internal control GAPDH.

## Drugs and reagents

Drugs were bath applied at the following concentrations: 6-cyano-7-nitroquinoxaline-2,3-dione (CNQX; 10 µM; Sigma; CAS: 115066-14-3), DL-2-amino-5-phosphonopentanoic acid (APV; 50 µM; Sigma; CAS: 76326-31-3) and gabazine (10 µM; Sigma; CAS: 104104-50-9), CsCl (3 mM; sigma; CAS: 7647-17-8), ZD7288 (60 µm; Abcam; CAS: 133059-99-1), TTX (1 µM; MCE and CHENGDU MUST BIO-TECHNOLOGY CO., LTD; CAS: 4368-28-9), GdCl$_3$ (100 µM; Sigma; CAS: 10138-52-0), L703,606 (10 µM; Sigma; CAS: 351351-06-9), 2-Aminoethyldiphenylborinate (2-APB, 100 µM; Sigma; CAS: 524-95-8), Larixyl acetate (10 µM; Sigma; CAS: 4608-49-5), Ruthenium Red (60 µM; TCI; CAS: 12790-48-6).

Commercial antibodies used were: Anti-Tyrosine Hydroxylase Antibody (1:400, Millipore, MAB318, RRID: AB_2201528, for Immunofluorescence), Anti-Dopamine Transporter (N-terminal) antibody (1:400, sigma, D6944, RRID: AB_1840807, for Immunofluorescence), Anti-NALCN (1:100, Thermo Fisher Scientific, MA5-27593, RRID: AB_2735285, for Immunofluorescence), Anti-TRPC6 (1:100, Alomone, ACC-017, RRID: AB_2040243, for Immunofluorescence), Anti-GAPDH (1:10000, Santa Cruz, sc-137179, RRID: AB_2232048, for Western Blot), Anti-NALCN (1:100, GeneTex, GTX54808, RRID: AB_3097701, for Western Blot), Anti-TRPC6 (1:100, Cell Signaling Technology, 16716, RRID: AB_2798768, for Western Blot).

Secondary antibodies: Donkey anti-Mouse IgG (H+L) Highly Cross-Adsorbed Secondary Antibody (Alexa Fluor 488, Thermo Fisher Scientific, A-21202, 1:1000, RRID: AB_141607), Donkey anti-Mouse IgG (H+L) Highly Cross-Adsorbed Secondary Antibody (Alexa Fluor 546, Thermo Fisher Scientific, A10037, 1:1000, RRID: AB_11180865), Donkey anti-Rabbit IgG (H+L) Highly Cross-Adsorbed Secondary Antibody (Alexa Fluor 488, Thermo Fisher Scientific, A21206, 1:1000, RRID: AB_2535792), Donkey anti- Rabbit IgG (H+L) Highly Cross-Adsorbed Secondary Antibody (Alexa Fluor 546, Thermo Fisher Scientific, A10040, 1:1000, RRID: AB_2534016), Donkey anti-Rabbit IgG (H+L) Highly Cross-Adsorbed Secondary Antibody (Alexa Fluor 647, Thermo Fisher Scientific, A31573, 1:1000, RRID: AB_2536183).

## Quantification and statistical analysis

Software such as GraphPad Prism6, OriginPro 8.0 (Origin Lab), and Adobe Illustrator CS6 were used for data analysis and image processing. All experimental data were expressed as mean ± standard error (mean ± S.E.M.). When the data were normally distributed, the difference between the two groups was statistically analyzed by two-sample T-test or paired-sample T-test; when the data were not normally distributed, the difference between two groups was statistically analyzed by Mann-Whitney U test or Wilcoxon matched-pairs signed rank test. $P<0.05$ was considered as a statistically significant difference between the two groups. For the data between multiple groups, when the data were normally distributed and there was no significant variance in homogeneity, one-way ANOVA with Dunnett's multiple comparisons test or two-way ANOVA with Sidak's multiple comparisons test was used; when the data were not normally distributed, Kruslal-Wallis-H test with Dunnett's multiple comparisons test was used.

## Acknowledgements

We would like to thank colleagues at the Core Facilities and Centers of Hebei Medical University for their technical assistance. This work was supported by the National Natural Science Foundation of China (81871075, 82071533) grants to HZ; National Natural Science Foundation of China (81870872, U21A20359) grants to XD; Science Fund for Creative Research Groups of Natural Science Foundation of Hebei Province (no. H2020206474); Basic Research Fund for Provincial Universities (JCYJ2021010), Natural Science Foundation of Hebei Province (H2023423065), Science and Technology Research Project of Higher Education Institutions in Hebei Province (QN2024159) and Scientific Research Project of Hebei Administration of Traditional Chinese Medicine (2024091) grants to WJ.

## Additional information

### Competing interests

Min Su: is affiliated with Yiling Pharmaceutical Company. The other authors declare that no competing interests exist.

### Funding

| Funder | Grant reference number | Author |
|---|---|---|
| National Natural Science Foundation of China | 81871075 | Hailin Zhang |
| National Natural Science Foundation of China | 82071533 | Hailin Zhang |
| National Natural Science Foundation of China | 81870872 | Xiaona Du |
| National Natural Science Foundation of China | U21A20359 | Xiaona Du |
| Science Fund for Creative Research Groups of Natural Science Foundation of Hebei Province | H2020206474 | Hailin Zhang |
| Natural Science Foundation of Hebei Province | H2023423065 | Jing Wang |
| Basic Research Fund for Provincial Universities | JCYJ2021010 | Jing Wang |
| Science and Technology Research Project of Higher Education Institutions in Hebei Province | QN2024159 | Jing Wang |

| Funder | Grant reference number | Author |
|---|---|---|
| Scientific Research Project of Hebei Administration of Traditional Chinese Medicine | 2024091 | Jing Wang |

The funders had no role in study design, data collection and interpretation, or the decision to submit the work for publication.

## Author contributions
Jing Wang, Data curation, Writing – original draft; Min Su, Ludi Zhang, Chenxu Niu, Chaoyi Li, Shuangzhu You, Yuqi Sang, Yongxue Zhang, Methodology; Dongmei Zhang, Data curation; Xiaona Du, Project administration; Hailin Zhang, Funding acquisition, Writing – review and editing

## Author ORCIDs
Jing Wang ⓘD https://orcid.org/0000-0001-6216-2919
Ludi Zhang ⓘD https://orcid.org/0000-0003-2061-4563
Hailin Zhang ⓘD https://orcid.org/0000-0002-9305-9508

## Ethics
All experiments were conducted in accordance with the guidelines of Animal Care and Use Committee of Hebei Medical University and approved by the Animal Ethics Committee of Hebei Medical University (Shijiazhuang, China), 2017013.

Reviewer #1 (Public Review): https://doi.org/10.7554/eLife.88319.4.sa1
Reviewer #2 (Public Review): https://doi.org/10.7554/eLife.88319.4.sa2
Reviewer #3 (Public Review): https://doi.org/10.7554/eLife.88319.4.sa3
Author response https://doi.org/10.7554/eLife.88319.4.sa4

# Additional files

## Supplementary files
• MDAR checklist

## Data availability
All data generated or analysed during this study are included in the manuscript and supporting files; source data files have been provided for all Figures.

The following datasets were generated:

| Author(s) | Year | Dataset title | Dataset URL | Database and Identifier |
|---|---|---|---|---|
| Zhang H, Wang J | 2024 | The ion channel mechanisms of the subthreshold inward depolarizing currents in the VTA dopaminergic neurons and their roles in the depression-like behavior | https://www.ncbi.nlm.nih.gov/geo/query/acc.cgi?acc=GSE276319 | NCBI Gene Expression Omnibus, GSE276319 |
| Wang J | 2024 | The ion channel mechanisms of the subthreshold inward depolarizing currents in the mice VTA dopaminergic neurons and their roles in the depression-like behavior | https://doi.org/10.5061/dryad.41ns1rnq5 | Dryad Digital Repository, 10.5061/dryad.41ns1rnq5 |

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
