## [Editor Report · eLife assessment]

This **important** study examined the mechanisms underlying reduced excitability of ventral tegmental area dopamine neurons in mice that underwent a chronic mild unpredictable stress treatment. The authors identify NALCN and TRPC6 channels as key mechanisms that regulate spontaneous firing of ventral tegmental area dopamine neurons and examined their roles in reduced firing in mice that underwent a chronic mild unpredictable stress treatment. The authors' conclusions on neurophysiological data are supported by multiple approaches and are **convincing**, although the relevance of the behavioral results to human depression remains unclear.

---

## [Referee Report · Reviewer #1 (Public Review)]

Wang et al., present a paper aiming to identify NALCN and TRPC6 channels as key mechanisms regulating VTA dopaminergic neuron spontaneous firing and investigating whether these mechanisms are disrupted in a chronic unpredictable stress model mouse.

Major strengths:

This paper uses multiple approaches to investigate the role of NALCN and TRPC6 channels in VTA dopaminergic neurons.

---

## [Referee Report · Reviewer #2 (Public Review)]

This paper describes the results of a set of complementary and convergent experiments aimed at describing roles for the non-selective cation channels NALCN and TRPC6 in mediating subthreshold inward depolarizing currents and action potential generation in VTA DA neurons under normal physiological conditions. In general, the authors have responded satisfactorily to reviewer comments, and the revised manuscript is improved.

---

## [Referee Report · Reviewer #3 (Public Review)]

The authors of this study have examined which cation channels specifically confer to ventral tegmental area dopaminergic neurones their autonomic (spontaneous) firing properties. Having brought evidence for the key role played by NALCN and TRPC6 channels therein, the authors aimed at measuring whether these channels play some role in so-called depression-like (but see below) behaviors triggered by chronic exposure to different stressors. Following evidence for a down-regulation of TRPC6 protein expression in ventral tegmental area dopaminergic cells of stressed animals, the authors provide evidence through viral expression protocols for a causal link between such a down-regulation and so-called depression-like behaviors. The main strength of this study lies on a comprehensive bottom-up approach ranging from patch-clamp recordings to behavioral tasks. These tasks mainly address anxiety-like behaviors and so-called depression-like behaviors (sucrose choice, forced swim test, tail suspension test). The results gathered by means of these procedures are clearcut.

---

## [Author Response]

The following is the authors’ response to the previous reviews.

**Reviewer #1 (Public Review):**
Comment 1: One of the only demonstrations of the expression and physiological significance of TRPCs in VTA DA neurons was published by (Rasmus et al., 2011; Klipec et al., 2016) which are not cited in this paper. In their study, TRPC4 expression was detected in a uniformly distributed subset of VTA DA neurons, and TRPC4 KO rats showed decreased VTA DA neuron tonic firing and deficits in cocaine reward and social behaviors. Update: The authors say they have added a discussion of these papers, but I do not see it in the updated manuscript.

We thank the reviewer for the suggestion. The discussion for this has been added (line 557-565).

Comment 2: The authors should report the results (exact data values) of female mice in the results text, or pool the male and female data if the sex differences are not significant.

We agree with reviewer. Some experiments were further redone with female and the data of male and female mice have been reported in the results of text.

Comment 3: The selectivity of drugs should be referred as "selective" rather than "specific".

Thanks, “specific” has been changed to “selective”.

Comment 4: Line 62: typo, "substantia nigra".

Thanks, “substantial nigra” has been changed to “substantia nigra” in line 65.

Comment 5: Line 77: some new studies suggest that NALCN might have voltage dependency

(rectification).

Thanks, description of NALCN voltage dependence has been corrected in line 81-83.

Comment 6: Line 175: change "less" to "fewer".

Thanks, “less” has been changed to “fewer”.

Comment 7: Line 299: choose one - "was not ... or" or "was neither ... nor".

Thanks, this error has been corrected.

Comment 8: In Figure 1Aii and Figure 3Bi, it was not specified in the results text or figure legend that C1-C5 represent individual cell until the legend for Figure 4.

Thanks, these description about gel have been added in the figure legends.

**Reviewer #2 (Public Review):**
Comment 1: From the previous review, we mentioned that " 'The HCN' as written in line 69 is a bit misleading, as HCN channels in the heart and brain are different members of a family of channels, although as written in the text, it seems that they are identical." This is still the case (now line 73).

We agreed with the reviewer’s comments. The introduction about HCN has been corrected (line 74-78).

Comment 2: The authors state in line 112 that "most of the experiments were also repeated in female mice" - this is true in the case of most electrophysiological experiments, although not behavioral experiments. Authors should amend the statement in line 112 and clarify in the Discussion section which findings are generalizable between sexes; e.g.:a. Discussion of HCN contribution to VTA DA activity (beginning line 453) should clarify male mice.b. Similarly, any discussion of behavioral findings should clarify male mice.

We agreed with the reviewer’s comments. The sexes of mice used have been noted in the results and discussion.

Comment 3: The authors' statement in lines 179-183 ("In contrast, fewer GABAergic neuronal markers (Glutamic acid decarboxylase, GAD1/2 and vesicular GABA transporter, VGAT) co-expressed with the DA neurons, which is consistent with previous studies that VTA DA neurons co-expressing GABAergic neuronal markers mainly project to the lateral habenula") is a little confusing - as stated, it seems that the authors are confirming DA/GABA coexpression in VTA-LHb neurons, which is not the case.

We agreed with the reviewer’s comments. We corrected this statement (line 182-186).

Comment 4: Additional information could be included in the Methods section description of Western Blotting procedures - e.g., what thickness of tissue and what size gauge were used to dissect VTA for these experiments?

Thanks. The description of tissue in Western Blotting procedures has been added.

Comment 5:a. Grammatical errors in line 23 of Abstract (also lines 31-32)b. "drove" should read "strove" in line 92c. Grammatical errors in lines 401, 444, and 448

We thank the reviewer for pointing out grammatical errors and we corrected them.

**Reviewer #3 (Public Review):**
Comment 1: The main strength of this study lies on a comprehensive bottom-up approach ranging from patch-clamp recordings to behavioral tasks. These tasks mainly address anxiety-like behaviors and so-called depression-like behaviors (sucrose choice, forced swim test, tail suspension test). The results gathered by means of these procedures are clearcut. However, the reviewer believes that the authors should be more cautious when interpreting immobility responses to stress (forced swim, tail suspension) as "depression-like" responses. These stress models have been routinely used (and validated) in the past to detect the antidepressant properties of compounds under investigation, which by no means indicates that these are depression models. For readers interested by this debate, I suggest to read e.g. De Kloet and Molendijk (Biol. Pscyhiatry 2021).

We thank the reviewer for the suggestion. We will be more careful and rigorous in the selection of stress models in our subsequent research work.

**Editor's note:**
Should you choose to revise your manuscript, please include full statistical reporting including exact p-values wherever possible alongside the summary statistics (test statistic and df) and 95% confidence intervals. These should be reported for all key questions and not only when the p-value is less than 0.05.

We have added the full statistical reporting including exact p-values wherever possible alongside the summary statistics (test statistic and df) and 95% confidence intervals into the results and the figure legends of the revised manuscript.